# Molecular and spatial transcriptomic classification of midbrain dopamine neurons and their alterations in a LRRK2[G2019S] model of Parkinson's disease

Zachary Gaertner[1,2,3†], Cameron Oram[4†], Amanda Schneeweis[1,3], Elan Schonfeld[1], Cyril Bolduc[4], Chuyu Chen[3,5], Daniel Dombeck[2,3], Loukia Parisiadou[3], Jean-Francois Poulin[4]*, Rajeshwar Awatramani[1,3]*

[1]Northwestern University Feinberg School of Medicine, Dept of Neurology, Chicago, United States; [2]Northwestern University, Dept of Neurobiology, Evanston, United States; [3]Aligning Science Across Parkinson's (ASAP) Collaborative Research Network, Chevy Chase, United States; [4]McGill University (Montreal Neurological Institute), Faculty of Medicine and Health Sciences, Dept of Neurology and Neurosurgery, Montreal, Canada; [5]Northwestern University Feinberg School of Medicine, Dept of Pharmacology, Chicago, United States

*For correspondence:
j-francois.poulin@mcgill.ca (J-FP);
r-awatramani@northwestern.edu (RA)

†These authors contributed equally to this work

Competing interest: The authors declare that no competing interests exist.

## eLife Assessment

This **important** study combines single nucleus transcriptional profiling with spatial transcriptomics to identify and map heterogeneity among dopamine neurons in the mouse ventral midbrain. The **compelling** results separate dopamine neurons into three broad families that have unique (yet overlapping) spatial distribution within the ventral tegmental area and substantia nigra, and also identify population-specific changes in a LRRK2 mouse model of Parkinson's Disease. The creation of a public-facing app where the snRNA-seq data can be investigated by anyone is a major strength.

**Abstract** Several studies have revealed that midbrain dopamine (DA) neurons, even within a single neuroanatomical area, display heterogeneous properties. In parallel, studies using singlecell profiling techniques have begun to cluster DA neurons into subtypes based on their molecular signatures. Recent work has shown that molecularly defined DA subtypes within the substantia nigra (SNc) display distinctive anatomic and functional properties, and differential vulnerability in Parkinson's disease (PD). Based on these provocative results, a granular understanding of these putative subtypes and their alterations in PD models, is imperative. We developed an optimized pipeline for single-nuclear RNA sequencing (snRNA-seq) and generated a high-resolution hierarchically organized map revealing 20 molecularly distinct DA neuron subtypes belonging to three main families. We integrated this data with spatial MERFISH technology to map, with high definition, the location of these subtypes in the mouse midbrain, revealing heterogeneity even within neuroanatomical sub-structures. Finally, we demonstrate that in the preclinical LRRK2[G2019S] knock-in mouse model of PD, subtype organization and proportions are preserved. Transcriptional alterations occur in many subtypes including those localized to the ventral tier SNc, where differential expression is observed in synaptic pathways, which might account for previously described DA release deficits in this model. Our work provides an advancement of current taxonomic schemes of the mouse midbrain DA neuron subtypes, a high-resolution view of their spatial locations, and their alterations in a prodromal mouse model of PD.

## Introduction

Midbrain DA neurons are traditionally organized in three main anatomical areas – the ventral tegmental area (VTA), SNc, and retrorubral area (RR) (*Björklund and Dunnett, 2007*), with further sub-divisions therein (*Fu et al., 2012*). To explain the disparate roles of DA in normal behavior and disease, additional heterogeneity of midbrain neurons within these areas has been postulated (*Gaertner et al., 2022*). Recent studies have indeed revealed diversity within clusters in electro-physiological properties as well as responses during various behavioral paradigms (*Evans et al., 2017*; *Menegas et al., 2018*; *Engelhard et al., 2019*; *Lammel et al., 2008*; *da Silva et al., 2018*; *Matsumoto and Hikosaka, 2009*; *Beier et al., 2015*; *Avvisati et al., 2024*; *Brischoux et al., 2009*; *Heymann et al., 2020*; *Howe and Dombeck, 2016*; *Lerner et al., 2015*). Complementing these studies, recent evidence using single-cell classification has opened the possibility that DA neurons can be clustered based on their molecular signatures (*Phillips et al., 2022*; *Poulin et al., 2020*; *Garritsen et al., 2023*; *La Manno et al., 2016*; *Kramer et al., 2018*; *Hook et al., 2018*; *Tiklová et al., 2019*; *Saunders et al., 2018*; *Yaghmaeian Salmani et al., 2024*; *Azcorra et al., 2023*). Early studies have begun to suggest that molecularly distinct DA populations may have distinctive anatomical projection patterns, as well as functionally distinct activity patterns (*Evans et al., 2017*; *Azcorra et al., 2023*; *Wu et al., 2019*; *Poulin et al., 2018*). For example, in the SNc, activity in a ventral Anxa1 + population is correlated to acceleration in mice on a treadmill, whereas activity in dorsal Calb1 + or lateral Vglut2 + populations is correlated to deceleration (*Azcorra et al., 2023*). These results suggest that molecularly defined DA neurons need to be considered as a key variable for electrophysiological and behavioral studies.

Based on these exciting results, and the fact that the functional interrogation of DA neurons is happening at rapid speed often agnostic to DA subpopulations, there is a clear need to define DA neuron subtypes with more granularity. Previous studies using single-cell sequencing in mice have been limited by the number of cells analyzed, the quality of cDNA library, as well as bias during isolation, thus providing an incomplete picture of DA neuron subtypes (*Poulin et al., 2020*; *La Manno et al., 2016*; *Kramer et al., 2018*; *Hook et al., 2018*; *Tiklová et al., 2019*; *Saunders et al., 2018*). Furthermore, the anatomical distribution of these subtypes in the midbrain remains only partially elucidated. Additionally, given the closely related nature of these subtypes, the stability of these subtypes across pathological conditions remains unclear. In other words, are some of these 'subtypes' a representation of cell state rather than cell type?

The unbiased study of the transcriptomic landscape of DA neuron subtypes in the midbrain can inform about the molecular drivers of selective dysfunction of DA neurons in PD. Pathogenic mutations which increase leucine-rich repeat kinase 2 (LRRK2) activity are one of the most common causes of autosomal dominant PD and clinically similar to sporadic cases (*Greggio et al., 2006*; *Tokars et al., 2022*; *Taymans et al., 2023*). Additionally, even in patients with idiopathic PD, LRRK2 kinase activity is increased in DA neurons (*Di Maio et al., 2018*), providing justification for studying LRRK2-driven mechanisms in DA neurons. Structural and functional synaptic changes are a recurring theme with LRRK2 mutations (*Khan et al., 2024*; *Matikainen-Ankney et al., 2016*; *Chen et al., 2020*). Accordingly, impaired DA transmission is observed with several LRRK2 mouse models including knock-in (KI) mouse lines expressing the most common LRRK2 mutation (G2019S) at physiological levels (*Xenias et al., 2022*; *Yue et al., 2015*; *Tozzi et al., 2018*), suggestive of functional synaptic deficits in dopaminergic circuits in the absence of overt DA neuron loss. Thus, LRRK2 KI mouse models are valuable for investigating early PD mechanisms in vulnerable DA neurons. However, despite the clinical relevance, the cell-autonomous functions of LRRK2 in nigral DA neurons are largely unknown.

snRNA-seq offers an avenue to begin to understand downstream transcriptomic alterations within DA neuron subtypes. We have used snRNA-seq to surveil DA neurons, which allows greater acquisition of DA numbers than whole-cell approaches and minimizes isolation bias (*Azcorra et al., 2023*). Here, we extend previous studies by optimizing our isolation protocol, enabling sequencing of large numbers of DA nuclei. We provide a high-resolution view of DA subtypes and an online portal to interrogate this dataset. We systematically map DA subtype distribution using MERFISH, providing unprecedented spatial resolution. Finally, in a LRRK2$^{G2019S}$ preclinical mouse model of PD, we reveal molecular pathways that are altered in locomotion-relevant DA neuron subtypes.

## Results

### A large snRNA-seq dataset of midbrain dopaminergic neurons reveals novel molecular subtypes

We performed snRNA-seq from dopaminergic nuclei (defined by Dat-Cre expression) in LRRK2[G2019S] heterozygous knock-in mutants and control littermates using a similar technique to our previously published dataset (*Azcorra et al., 2023*) but modified to utilize a chip-based microfluidic sorting method which minimizes nuclear stress. This improved protocol enabled increased yield and sample quality (*Figure 1A*, *Figure 1—figure supplement 1A*). Doing so enabled us to generate a dataset with 28,532 profiled cells (after quality control filtering) with a median UMI count of 7750.5 and median of 3056 genes. Downstream clustering resulted in 22 unique clusters (*Figure 1B*), of which two clusters (16 and 21, 911 total cells) were determined to represent possible dopaminergic/glial doublets based on co-expression of glial markers in these populations and were excluded from downstream analyses (shown in gray in *Figure 1B*, *Figure 1—figure supplement 1F*). The 20 remaining clusters all showed expression of key pan-dopaminergic markers including *Slc18a2* (*Vmat2*), *Ddc*, *Th*, and *Slc6a3* (*Dat*) (*Figure 1C*). These clusters were observed in both males and females, and across genotypes, in similar proportions (*Figure 1—figure supplement 1B–E*). These subtypes showed distinct expression signatures of many previously described markers of DA neuron subtypes (*Figure 1D*; *Poulin et al., 2020*). However, our increased yield and sample quality also allowed explicit detection of known populations that have eluded most prior single-cell studies, such as DA neurons expressing *Vip* (*Poulin et al., 2020*; *Poulin et al., 2014*), thereby demonstrating the strength of this pipeline in detecting rare DA neuron subtypes. Importantly, all the described subtypes showed high expression of midbrain floorplate markers (*Figure 1E*), with exception of clusters 0 and 17, which possibly represent *Slc6a3+* neurons in the supramammillary and/or premamillary regions (*Soden et al., 2016*; *Nouri and Awatramani, 2017*). Thus, our increased number of subtypes over prior studies represents increased granularity among classic midbrain DA neuron subtypes, rather than detection of non-floorplate derived populations that express DA markers.

### Analysis of differentially expressed genes reveals families of subtypes and relationships to prior studies

Given the large number of clusters discovered in our dataset, we first sought to organize these subtypes into groups in an unbiased manner. By creating a cluster dendrogram (*Figure 2A*), we were able to map an approximation of the relative relatedness of different subtypes, which segregated into three main families based on major branch points. By running differential expression on each node of the cluster dendrogram (i.e. exploring differentially expressed genes (DEGs) between the two arms emerging from a given node), we were then able to create a stepwise expression code that can uniquely define any individual cluster (*Figure 2A and G*). The earliest division point among our clusters was defined by genes *Prkcd* and *Lef1*, which are highly expressed only in cluster 17 (Log2 fold changes = 6.279 and 7.246 respectively, BH-corrected p-values ≈ 0; note, such p-values are calculated using default Seurat differential expression functions which treat each cell as an independent sample thus potentially inflating statistical significance). Next, a family of clusters defined by *Gad2/Kctd16* (deemed Gad2 family, Log2 fold changes = 5.318 and 4.014 respectively, BH-corrected p-values ≈ 0) separated from all other clusters, which expressed significantly higher levels of *Ddc* and *Slc6a3* (Log2 fold changes = 1.419 and 1.643 respectively, BH-corrected p-values ≈ 0). The first major branch point between *Ddc*-high/*Slc6a3*-high clusters was then defined by cells expressing *Sox6/Col25a1* (Log2 fold changes = 2.720 and 2.305 respectively, BH-corrected p-values ≈ 0) versus those expressing *Calb1/Ndst3* (Log2 fold changes = 2.025 and 3.368 respectively, BH-corrected p-values ≈ 0), thus creating the Sox6 family and Calb1 families, respectively. This division between *Sox6* and *Calb1* is consistent with our previous work (*Poulin et al., 2014*), as well as a prior large single-cell profiling study that found this division to be the central branch point among DA neurons across several species including humans (*Kamath et al., 2022*; *Siletti et al., 2023*). Cluster divisions can be visualized by plotting the co-expression of DEGs taken from the branch points labeled in red (*Figure 2C–F*) on the UMAP plot. Of note, while these branch point markers describe the overall trends of gene expression, these genes are not perfectly distinct or universal among the members of these cluster families. For example, clusters 1, 11, and 13 show expression of both *Sox6* and

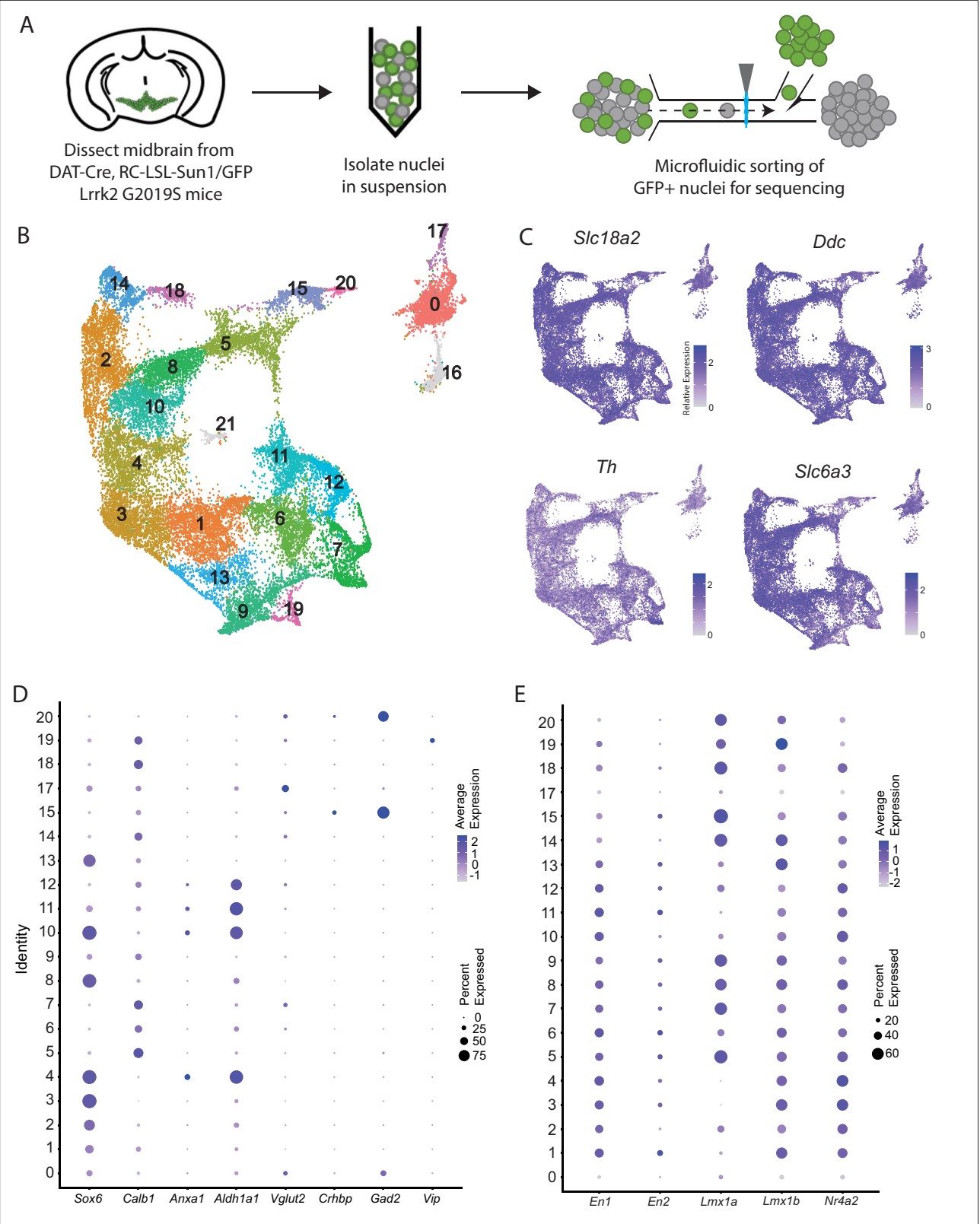

**Figure 1.** Generation of a large RNAseq dataset of midbrain dopamine neurons. (**A**) Schematic of single nucleus isolation pipeline using microfluidic chip-based sorting from n=8 DAT-IRES-Cre (referred to as DAT-Cre), CAG-Sun1/sfGFP (referred to as RC-LSL-Sun1/GFP) mice. (**B**) UMAP representation of clusters. Of note, putative clusters 16 and 21 are shown in gray, and are removed from analyses due to co-expression of glial markers. (**C**) Expression of the defining midbrain dopaminergic neuron markers vesicular monoamine transporter 2 (*Vmat2; Slc18a2*), DOPA decarboxylase (*Ddc*), tyrosine

*Figure 1 continued on next page*

Figure 1 continued

hydroxylase (*Th*), and dopamine transporter (*Dat; Slc6a3*). Robust expression is observed throughout all clusters. (**D**) Dot plot showing expression of several markers of previously described dopamine neuron subtypes, consistent with finding additional heterogeneity within established subtypes. (**E**) Expression patterns of midbrain floorplate markers. All clusters show robust expression with the exception of 0 and 17, which indicates these populations are not midbrain dopamine (DA) neurons but might be located in nearby hypothalamic nuclei.

The online version of this article includes the following figure supplement(s) for figure 1:

**Figure supplement 1.** Quality control for dataset generation.

*Calb1* (*Figure 1D*). The co-expression of these markers has previously been described in a fraction of mouse DA neurons (*Poulin et al., 2014*; *Anderegg et al., 2015*; *Pereira Luppi et al., 2021*). Furthermore, because the calculations of differential expression at each node take into account only the descendants of that branch point, markers defining a branch point may also be strongly expressed in clusters outside this comparison. Thus, while the final marker(s) listed for each subtype in *Figure 2A* are not unique to that population, following the sequential branches leading to this marker in a stepwise fashion will specifically enrich for this population (*Figure 2G*). By doing so, we provide a strategy for potential genetic access to any individual DA subtype using intersectional logic gates.

To enable comparisons of DA populations across existing and future literature, we next sought to better define genetic signatures for each of the individual putative subtypes and to provide a common language for discussing these populations. To do so, we utilized two complementary approaches. First, we explored the top differentially expressed genes with positive relative expression in each cluster (*Figure 2B*). While many of these markers are expressed in more than one population, some (e.g. *Vip*) are almost entirely unique to a single cluster. As a complementary approach, we performed differential expression among clusters within each of our three subtype families, thereby creating a shorthand identity based on the cluster family and top DEGs (*Figure 2G*). Within the Sox6 family, these clusters are Sox6 [Tmem132d] (listed as cluster 1 in *Figures 1 and 2*), Sox6[Arhgap28] (cluster 2), Sox6[March3] (cluster 3), Sox6[Tafa1] (cluster 4), Sox6[Kcnmb2] (cluster 8), and Sox6[Vcan] (cluster 10); in the Calb1 family, these are Calb1[Pde11a] (cluster 5), Calb1[Kctd8] (cluster 6), Calb1[Ptprt] (cluster 7), Calb1[Sulf1] (cluster 9), Calb1[Stac] (cluster 11), Calb1[Chrm2] (cluster 12), Calb1[Sox6] (cluster 13), Calb1[Lpar1] (cluster 14), Calb1[Ccdc192] (cluster 18), Calb1[Gipr] (cluster 19); in the Gad2 family, these are Gad2[Syndig1] (cluster 0), Gad2[Egfr] (cluster 15), and Gad2[Ebf2] (cluster 20). Cluster 17 (defined by Lef1) did not have an associated cluster family (*Figure 2A*). These shorthand identities provide a simple nomenclature for our clusters, which we have utilized throughout the remainder of this paper.

After establishing genetic signatures for each putative subtype, we were next able to correlate our cluster identities to those described in another recent study (*Azcorra et al., 2023*), as well as the consensus subtypes proposed in *Figure 2G*; *Poulin et al., 2020*. Doing so provides context for discussing our putative DA neuron subtypes with regards to prior literature. For example, cluster Sox6[Tafa1] was found to be equivalent to a population we previously defined by high expression of *Anxa1*, which holds notably distinct functional and anatomical properties (*Azcorra et al., 2023*). Several of the subtypes that emerged in our new dataset also appear to be novel divisions within clusters that previously showed evidence of internal heterogeneity based on their cluster stability metrics (*Azcorra et al., 2023*). For instance, a *Vip* + cluster (Calb1[Gipr]) emerged from within a highly variable cluster (defined as #10 in *Azcorra et al., 2023*), and additional subdivisions emerged from previously described unstable Sox6+ clusters (most notably #3 in *Azcorra et al., 2023*; *Figure 2G*). This emergence of unique populations, such as Calb1[Gipr], is likely due to the increased yield and sample quality of our new dataset which has enabled better detection of small populations and gene markers with lower expression levels. This is further supported by the high cluster stability of the new Calb1[Gipr] (*Figure 2—figure supplement 1A and B*), resolving the instability of the equivalent cluster in *Azcorra et al., 2023*. We re-clustered our snRNA-seq data into the clusters defined in *Azcorra et al., 2023* and found our clusters aligned with or further subdivided many of the previously defined clusters (*Figure 2—figure supplement 2A*). Lastly, to further corroborate our cluster identities, we used an independent bioinformatic platform, Scanpy, to re-cluster our nuclei (*Figure 2—figure supplement 2B*). Comparing the Seurat and Scanpy clusters using a Sankey diagram (*Figure 2—figure supplement 2C*) shows a high correlation between the Seurat and Scanpy clusters.

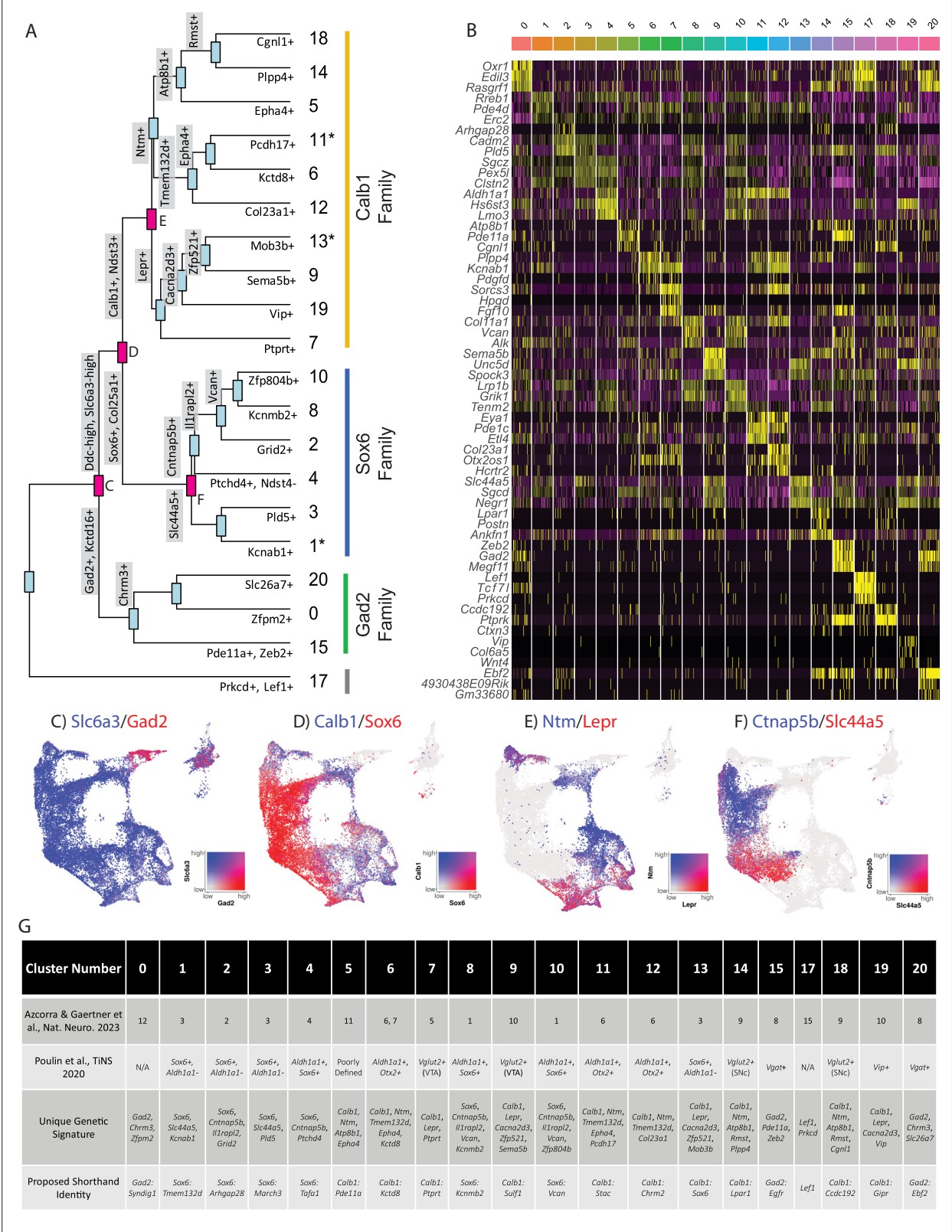

**Figure 2.** Mapping subtype identities, marker genes, and relationships to previously described populations. (**A**) Dendrogram of dopamine (DA) neuron subtypes in our dataset. Distance between branch points represents approximations of relatedness between subsequent nodes, and genes labeled at each branch point represent the top differentially expressed genes distinguishing the downstream groups. Subtypes fell largely into three 'families,' with expression patterns generally defined by *Gad2*, *Sox6*, or *Calb1*. Notably, clusters 1, 11, and 13 are shown with an asterisk as these do not display

*Figure 2 continued on next page*

*Figure 2 continued*

differential expression in line with the defining genes of their cluster families (i.e. are not significantly enriched for their eponymous family genes). (**B**) Heatmap of top differentially expressed genes for each cluster. The top three genes for each cluster are shown. (**C–F**) Co-expression of genes at specified branch points denoted in 2 A. (**G**) Table describing putative clusters and relation to previous literature. Top row: cluster number. Second row: Equivalent cluster in *Azcorra et al., 2023*. Third row: stepwise genetic signatures based on the following sequential branch points of cluster dendrogram. Bottom row: Proposed shorthand name for subtypes, based on cluster family and top defining genes for that cluster within each family.

The online version of this article includes the following figure supplement(s) for figure 2:

**Figure supplement 1.** Representations of cluster heterogeneity.

**Figure supplement 2.** Validating single-nuclear RNA sequencing (snRNA-seq) clusters with Python and published dopamine (DA) clusters.

## Mapping the spatial location of midbrain DA neurons using MERFISH

Having identified multiple clusters of midbrain DA neurons, we next resolved to determine the spatial distribution of these populations using MERFISH (*Chen et al., 2015*), a form of spatial transcriptomics that offers subcellular resolution imaging-based RNA quantification. We assembled a 500 gene panel which included 120 of the top DEGs identified within the snRNA-seq dataset, as well as 380 general markers of neuronal and non-neuronal cell populations (*Supplementary file 1*). Seven coronal slices were collected from adult mouse midbrains and processed for MERFISH (*Figure 3A*). The coronal slices were selected containing ventral midbrain regions between –2.9 mm to –3.9 mm from bregma. Cells were segmented using the CellPose algorithm based on DAPI and Poly-T staining and passed through quality control filters to remove cells with too large of a volume (potential doublets) or too few transcripts detected (false cells; *Figure 3—figure supplement 1A–C*). This resulted in 429,713 identified cells (*Figure 3B*). Cells from the seven sections displayed significantly correlated gene expression and minimal batch effects (*Figure 3—figure supplement 1D*). Cells were integrated in Seurat using reciprocal principal component analysis (RPCA) algorithm and then clustered, resulting in 40 final clusters comprising distinct classes of neurons, oligodendrocytes, oligodendrocyte precursor cells (OPCs), astrocytes, microglia, and meningeal cells (*Figure 3B*). These clusters could be attributed to 25 functional groups based on the expression of cell type markers from *Supplementary file 1* and their spatial distribution. Glia-associated genes were mostly contained within their expected cell types: oligodendrocytes (*Mog*, *Cnp*), OPC (*Pgfra*, *Olig2*), astrocytes (*Aqp4*, *Gfap*), and microglia (*Aif1*, *Tmem119*). However, we noted the unexpected distribution of the oligodendrocyte-specific transcript *Mbp* in non-glial cells which we believe might indicate the presence of this mRNA in glial-processes that overlap non-glial cells (*Bradl and Lassmann, 2010*). Within the neuronal class, we identified 16 excitatory clusters (presence of *Slc17a6* or *Slc17a7*), 7 inhibitory clusters (presence of *Slc32a1*, *Gad1* or *Gad2*), one serotonin neuron cluster, and one DA neuron cluster (*Figure 3B-D*, *Figure 3—figure supplement 1G*). Although our gene panel was designed to identify neuronal heterogeneity of the midbrain, it had sufficient resolution to delineate seven excitatory neuron classes present in distinct layers of the cortex and six cortical interneuron classes (*Figure 3—figure supplement 1E, F*). Cortical excitatory neurons were absent from subcortical structures such as the thalamus, hypothalamus, midbrain, and hindbrain. This fundamental dichotomy between cortical and subcortical neurons in the adult mouse brain is supported by recent spatial transcriptomic studies (*Zhang et al., 2023*; *Yao et al., 2023*; *Langlieb et al., 2023*).

A population of DA neurons was identified among our 40 clusters based on the expression *Th*, *Slc6a3*, and *Slc18a2* (*Figure 3C*). The location of the dopaminergic cluster (blue in *Figure 3D*) considerably overlaps with *Th* expression (*Figure 3E*). This cluster comprised 4532 cells, was entirely located in the midbrain, and was also enriched for *Ddc*, *Dlk1*, and *Drd2* (not shown). Within this population, we detected expression of genes associated with glutamatergic and GABAergic neurotransmission, in line with previous reports of multi-lingual characteristics of DA neurons (*Trudeau et al., 2014*; *Descarries et al., 2008*; *Morales and Root, 2014*; *Morales and Margolis, 2017*; *Tritsch et al., 2016*; *Conrad et al., 2024*), for instance, the glutamatergic marker *Slc17a6* (Vglut2) and GABAergic markers *Gad2* and *Slc32a1* (Vgat) (*Figure 3—figure supplement 1G*). Markers for cholinergic (*Chat*) and serotonergic (*Tph2*) neurotransmission were not detected in putative DA neurons and the glutamatergic transporters *Slc17a7* (Vglut1) and *Slc17a8* (Vglut3) were also absent. We next wanted to characterize the neuroanatomical location of this DA neuron cluster. We manually delineated neuroanatomical boundaries on each of the slices for seven midbrain regions known to contain most dopamine neurons

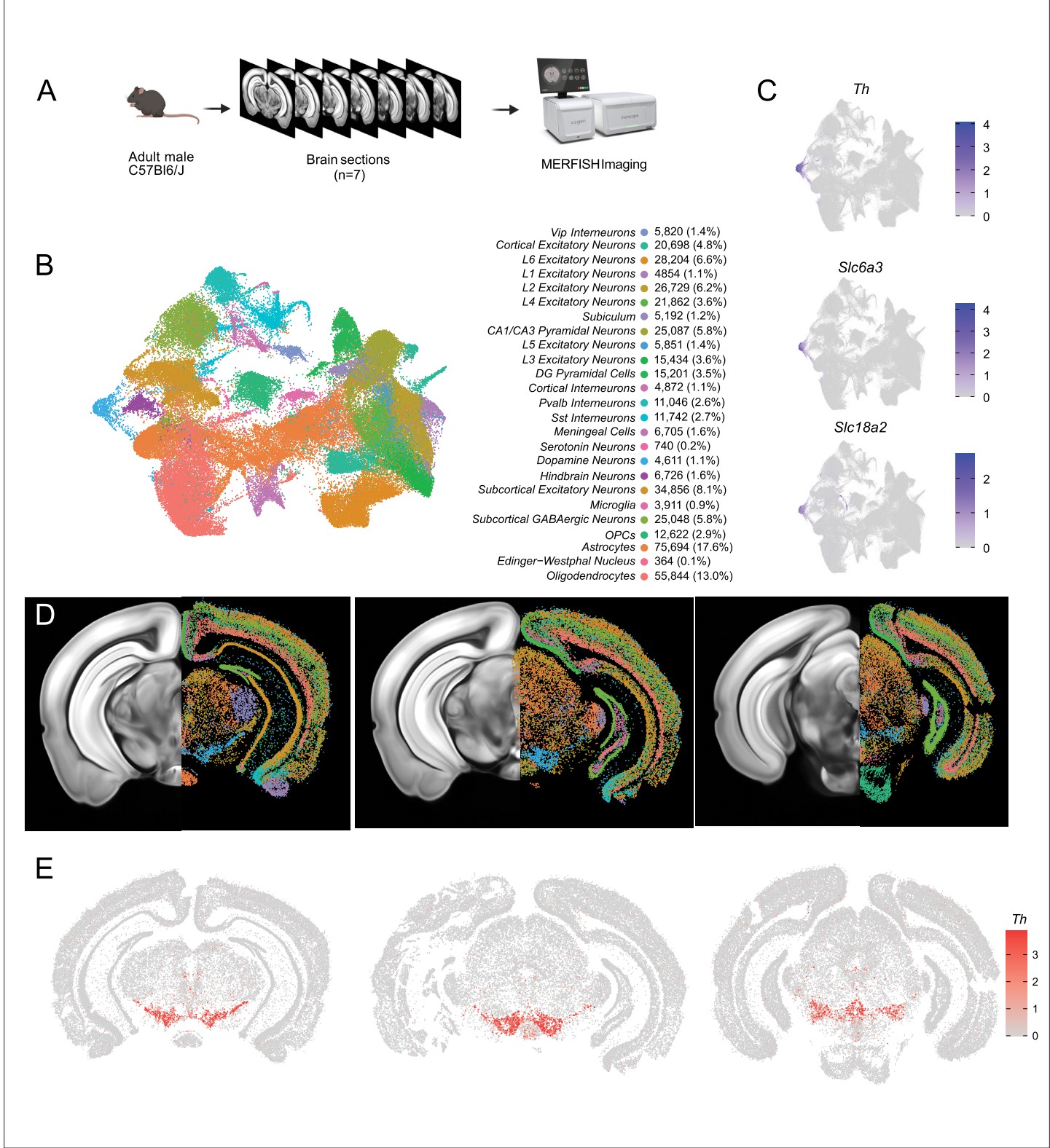

**Figure 3.** Identification of dopamine neurons in MERFISH. (**A**) Schematic showing the processing and imaging of brain tissue with MERFISH. (**B**) Clustering and annotation of all cells identified by MERFISH. (**C**) Relative expression of *Th*, *Slc6a3*, and *Slc18a2* shows the presence of a single cluster (blue cluster in B) that expresses all three genes. (**D**) Spatial location of neuronal clusters for whole brain MERFISH along three rostral-caudal delineations associated with −3.0, −3.2, and −3.6 mm Bregma. (**E**) Cellular expression of *Th* in the sections shown in (**D**).

*Figure 3 continued on next page*

*Figure 3 continued*

The online version of this article includes the following figure supplement(s) for figure 3:

**Figure supplement 1.** Quality control metrics for MERFISH experiments.

**Figure supplement 2.** Clustering of dopamine (DA) neurons identified by MERFISH.

including SNc, VTA, RR, caudal linear nucleus (CLi), interfascicular nucleus (IF), rostral linear nucleus (RLi), and periaqueductal gray (PAG). These boundaries were delineated based on cell density (dapi), white matter landmarks, and *Th* transcript distribution using the Allen Brain Atlas as a reference (*Figure 3—figure supplement 2A*). We found that cells in our DA neuron cluster were located in the following locations: 35.2% in the SNc, 19.8% in the VTA, 7.8% in the CLi, 15.3% in the RR, 2.0% in the IF, 0.6% in the RLi, 4% in the PAG and 15.3% located elsewhere (*Figure 3—figure supplement 2B*). Finally, to explore the diversity of DA neurons identified by MERFISH, we subclustered the 4532 cells to resolve 12 distinct subgroups (*Figure 3—figure supplement 2C–F*). For clarity, we designated the clusters identified solely with MERFISH transcript levels as MER-0–11. Many of the clusters were characterized by increased expression of known subtype markers such as *Calb1* (MER-6, MER-8 and MER-10), *Aldh1a1* (MER-5), *Ndnf* (MER-0), *Gad2* (MER-7 and MER-9), and *Slc17a6* (MER-2).

## Cell label transfer allows the mapping of snRNA-seq clusters

To provide the spatial distribution of the clusters identified using snRNA-seq, we sought to project the MERFISH dataset onto the snRNA-seq UMAP space. We first compared the expression level of the 500 genes utilized in our panel to select features for integration with comparable expression levels. We selected 281 genes to be included based on similarity in the average counts per cell between the datasets. Some excluded genes potentially reflected faulty probes or improper assignment of RNA transcripts to segmented cells. For instance, *Sox6* transcripts were not detected across the entire brain (data not shown). We successfully projected the MERFISH dataset onto the snRNA-seq UMAP space and selected for 2297 dopamine neurons with a cell similarity score >0.5 (mean value, 0.538), increasing confidence in the correct assignment to snRNA-seq defined clusters. Almost all clusters were represented by more than 20 cells in our MERFISH dataset (*Figure 4A*, *Figure 4—figure supplement 1A*). Clusters Calb1^Ccdc192 and Calb1^Gipr were not resolved in this label transfer either due to their relatively smaller number or location in an under-sampled region. Indeed, the *Vip*-expressing Calb1^Gipr has been shown to be located in the caudal midbrain (PAG/DR) (*Poulin et al., 2014*), a region not well covered in our MERFISH experiment. Comparing snRNA-seq clusters with the ones obtained by clustering the MERFISH dataset alone showed a strong correspondence (*Figure 4—figure supplement 1B*). Half of the snRNA-seq clusters were predominantly comprised of single MERFISH clusters (10/20) and 25% (5/20) were comprised of two MERFISH clusters, demonstrating that the integration with the snRNA-seq dataset allowed us to further refine our MERFISH-based classification (*Figure 4—figure supplement 1B*). We then imputed expression data from the snRNA-seq dataset onto the MERFISH cells to better visualize spatial expression levels. The imputed gene expression is extrapolated from anchors established from pairwise correspondences of cell expression levels between MERFISH and snRNA-seq datasets. The imputed data across the genes used for data integration correlated strongly with MERFISH transcripts, particularly for *Calb1*, *Gad2,* and *Aldh1a1* genes (*Figure 4—figure supplement 1C-F*, *Figure 4—figure supplement 2*). Altogether, the dataset integration allowed us to map the neuroanatomical location of 18 of the clusters identified by snRNA-seq as well as impute expression of the whole transcriptome.

## Locating DA neuron clusters within the midbrain

Clusters were sorted into Gad2, Sox6, and Calb1 families based on our hierarchical dendrogram (*Figure 4A*; *Gad2* n=124 cells, *Sox6* n=887 cells, *Calb1* n=590 cells). Imputed *Gad2*, *Sox6*, and *Calb1* gene expression within DA neurons showed a distinct medio-lateral and rostro-caudal distribution (*Figure 4C–E*). The spatial distribution of imputed *Sox6* and *Calb1* transcripts within DA neurons correlated well with previous reports (*Poulin et al., 2014*; *Pereira Luppi et al., 2021*; *Panman et al., 2014*) whereas the distribution of the Gad2 family is predominantly observed in the rostral linear/ posterior hypothalamus region and CLi, and to a lesser extent in the VTA (*Figure 4B*, *Figure 4— figure supplement 3*). Each family was overrepresented in a neuroanatomically-defined area; for

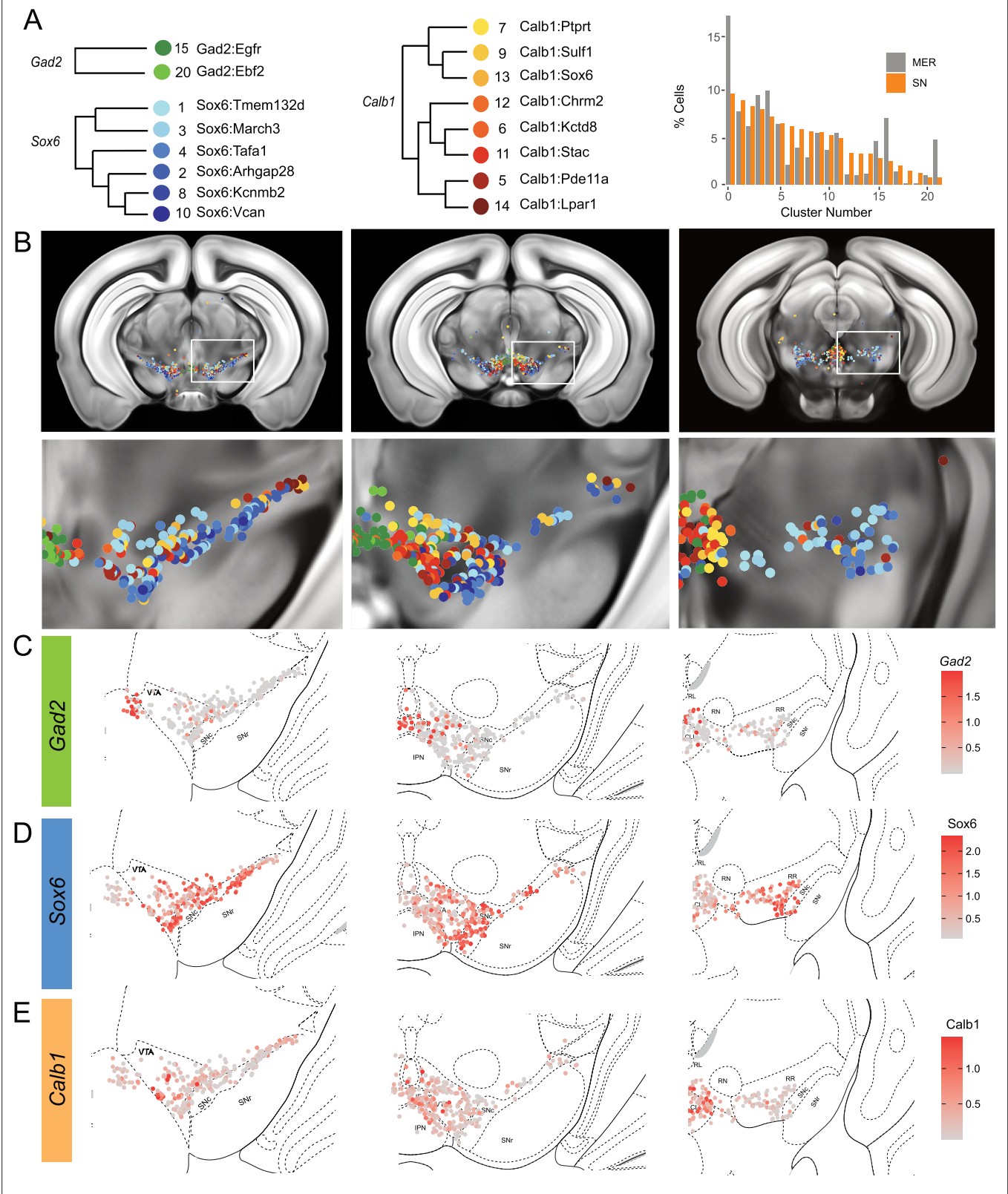

**Figure 4.** Mapping of MERFISH data with single-nuclear RNA sequencing (snRNA-seq) clustering: (**A**) Grouping and recoloring of snRNA-seq clusters into distinct families represented by Gad2, Sox6, and Calb1 (Left). Relative proportion of dopamine neurons comprising each cluster from the snRNA-seq (SN, orange) and MERFISH label transfer (MER, gray) data (Right). (**B**) Spatial representation of predicted clustering across the rostral-caudal axis, zoom of one representative hemisphere shown below. (**C–E**) Imputed cellular expression of *Gad2*, *Calb1*, and *Sox6* across the rostral-caudal axis.

*Figure 4 continued on next page*

*Figure 4 continued*

The online version of this article includes the following figure supplement(s) for figure 4:

**Figure supplement 1.** Integrating MERFISH with single-nuclear RNA sequencing (snRNA-seq): (**A**) Overview of MERFISH integration with snRNA-seq dataset (top).

**Figure supplement 2.** Correlation of MERFISH vs imputed transcript counts for markers of the Sox6 (**A**) and Calb1 (**B**) families.

**Figure supplement 3.** Quantifications of subtypes by anatomic localization.

instance, the SNc is composed of 78.4% of neurons from the Sox6 family (*Figure 4—figure supplement 3*), consistent with previous reports (*Pereira Luppi et al., 2021*). However, each family was also found across multiple neuroanatomical regions (*Figure 4—figure supplement 3A*). We also observed spatial heterogeneity within families. For example, one branch of the Calb1 family dendrogram (*Figure 4*, label in shades of yellow-orange defined by higher *Lepr* expression) was enriched in the dorsal VTA, whereas the other Calb1 branch (*Figure 4*, label in shades of red defined by higher *Ntm* expression) populated mostly the ventromedial VTA. Similar observations could be made for the Sox6 family, where subtypes displayed biased distributions. This led us to explore the distribution of each individual cluster and representative markers toward the goal of generating a granular map of the midbrain DA system.

The hierarchical clustering of the Sox6 family reveals two major branches (*Figures 2A and 4A*): (1) Branch 1 defined by higher levels of *Slc44a5* and composed of subtypes Sox6$^{Tmem132d}$ and Sox6$^{March3}$ and (2) Branch 2 defined by *Cntnap5* and composed of subtypes Sox6$^{Tafa1}$, Sox6$^{Arhgap28}$, Sox6$^{Kcnmb2}$, and Sox6$^{Vcan}$. We observed an important discrepancy between these two branches, with Branch 1 being located more dorsal and medial than Branch 2 (*Figure 5A*). Interestingly, this dorsoventral organization breaks down in the caudal SNc and RR where neurons from both branches are mostly intermixed. Only minor differences in location were observed between clusters of Branch 1, maybe reflecting their molecular similarities. Of note, whereas Sox6$^{Tmem132d}$ is predominant in the rostral SNc and VTA for this branch, neurons of Sox6$^{March3}$ are more numerous in the RR (*Figure 5B*, *Figure 4—figure supplement 3B*). Within Branch 2, the molecular identity correlated with the medio-lateral distribution. For instance, neurons of cluster Sox6$^{Arhgap28}$ were found to be more lateral in the SNc compared to neurons of Sox6$^{Tafa1}$, Sox6$^{Kcnmb2}$, or Sox6$^{Vcan}$. This medio-lateral distribution of neurons of the SNc ventral tier might reflect the differential topographical distribution of nigrostriatal projections previously reported (*Wu et al., 2019*; *Poulin et al., 2018*).

The Calb1 family was the most molecularly diverse which was also reflected in its localization (*Figure 6*). Two major branches of Calb1 DA neurons were identified: (1) Branch 1 (*Figure 6A–B*) containing subtypes Calb1$^{Ptprt}$, Calb1$^{Sulf1}$, and Calb1$^{Sox6}$ and (2) Branch 2 (*Figure 6C–D*) containing subtypes Calb1$^{Chrm2}$, Calb1$^{Kctd8}$, Calb1$^{Stac}$, Calb1$^{Pde11a}$, and Calb1$^{Lpar1}$. Broadly speaking, Calb1 subtypes were predominantly localized to the VTA and CLi with some neurons showing localization to the dorsal SNc, SNpl, or RR (*Figure 4—figure supplement 3*). Within the VTA, Branch 1 Calb1 neurons tended to localize to the dorsal VTA (*Figure 6A–B*) while Branch 2 Calb1 neurons showed distinct ventromedial localization (*Figure 6C–D*). In more caudal sections the difference was even more pronounced, although intermingling was always observed. Within Branch 1, subtypes Calb1$^{Ptprt}$ and Calb1$^{Sox6}$ were largely absent from the SNc dorsal tier and RR, whereas subtype Calb1$^{Sulf1}$ was present. Some notable distinctions were observed in both groups, specifically subtypes Calb1$^{Sulf1}$ and Calb1$^{Lpar1}$ which were each observed in significant numbers in the dorsolateral SNc/SNpl region. Of these, only Calb1$^{Lpar1}$ expressed significant *Slc17a6* (data not shown), making it a likely candidate for the SNpl Vglut2 + DA neuron population found to innervate the tail of the striatum (*Poulin et al., 2018*). Indeed, these laterally located neurons have been genetically targeted with a Slc17a6-Cre driver, and display deceleration-correlated responses, and robust responses to physically aversive stimuli (*Azcorra et al., 2023*; *Poulin et al., 2018*).

Within the Gad2 family, we were able to resolve two subtypes. Subtypes Gad2$^{Egfr}$ and Gad2$^{Ebf2}$ had very similar spatial distributions and were prominent in the posterior IF, RLi, and CLi (*Figure 7*, *Figure 4—figure supplement 3*). These regions have been associated with small DA neurons of lower *Th* and *Slc6a3* levels. Subtype Gad2$^{Egfr}$ neurons were more plentiful than cluster Gad2$^{Ebf2}$, with the latter being absent in more caudal sections. All Gad2 subtypes expressed *Slc32a1*, opening the

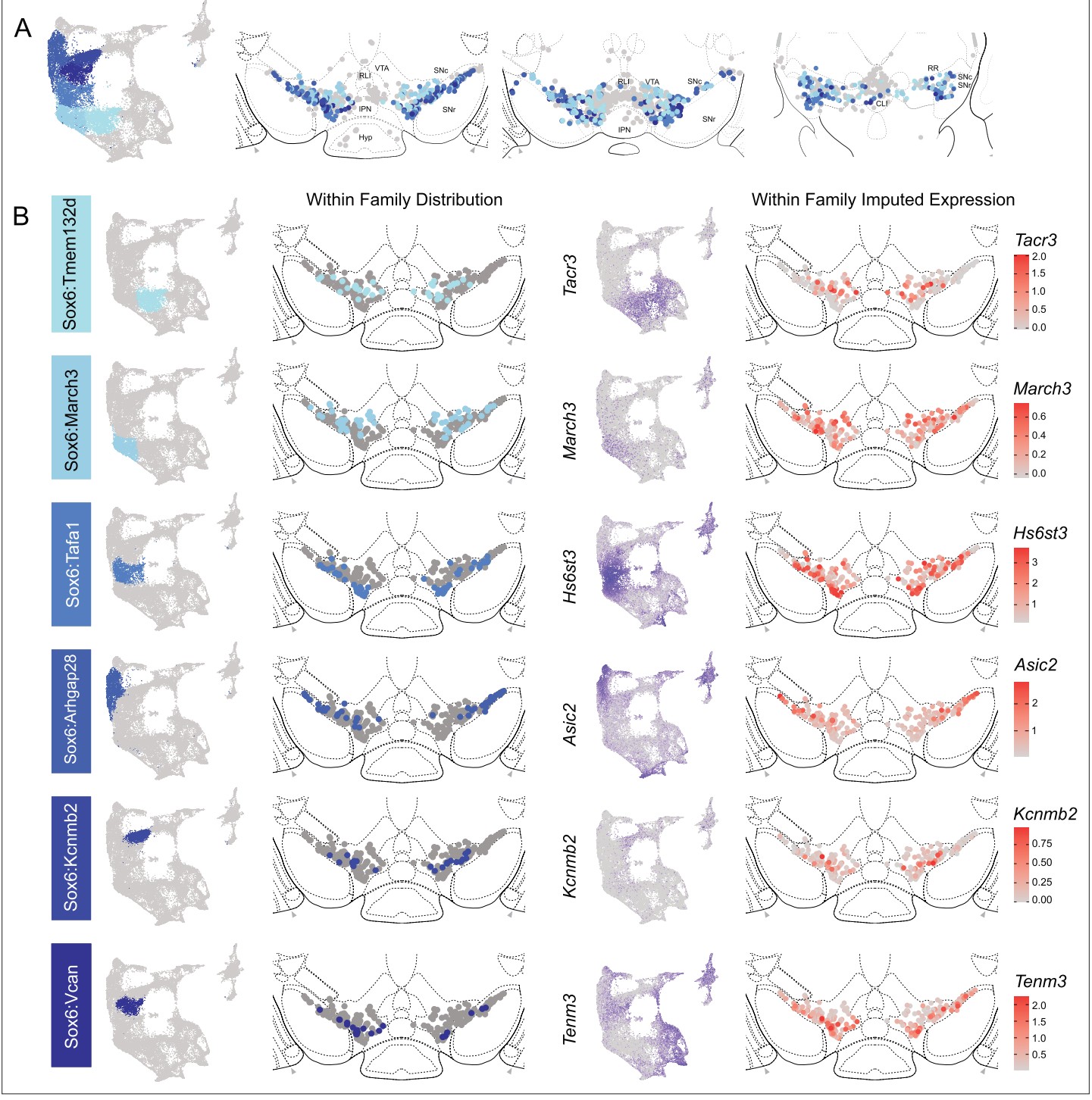

**Figure 5.** Spatial localization of Sox6 family of dopamine (DA) neurons. (**A**) Distribution of Sox6 family DA neuron subtypes relative to all other DA neurons along the rostral-caudal axis. DA cells belonging to the Calb1 and Gad2 families are colored in light gray. (**B**) Location of individual subtype within the Sox6 family (left two panels) with other cells in the Sox6 family shown in dark gray. Relative cellular expression of genetic markers associated with each subfamily is shown in a UMAP and a midbrain section.

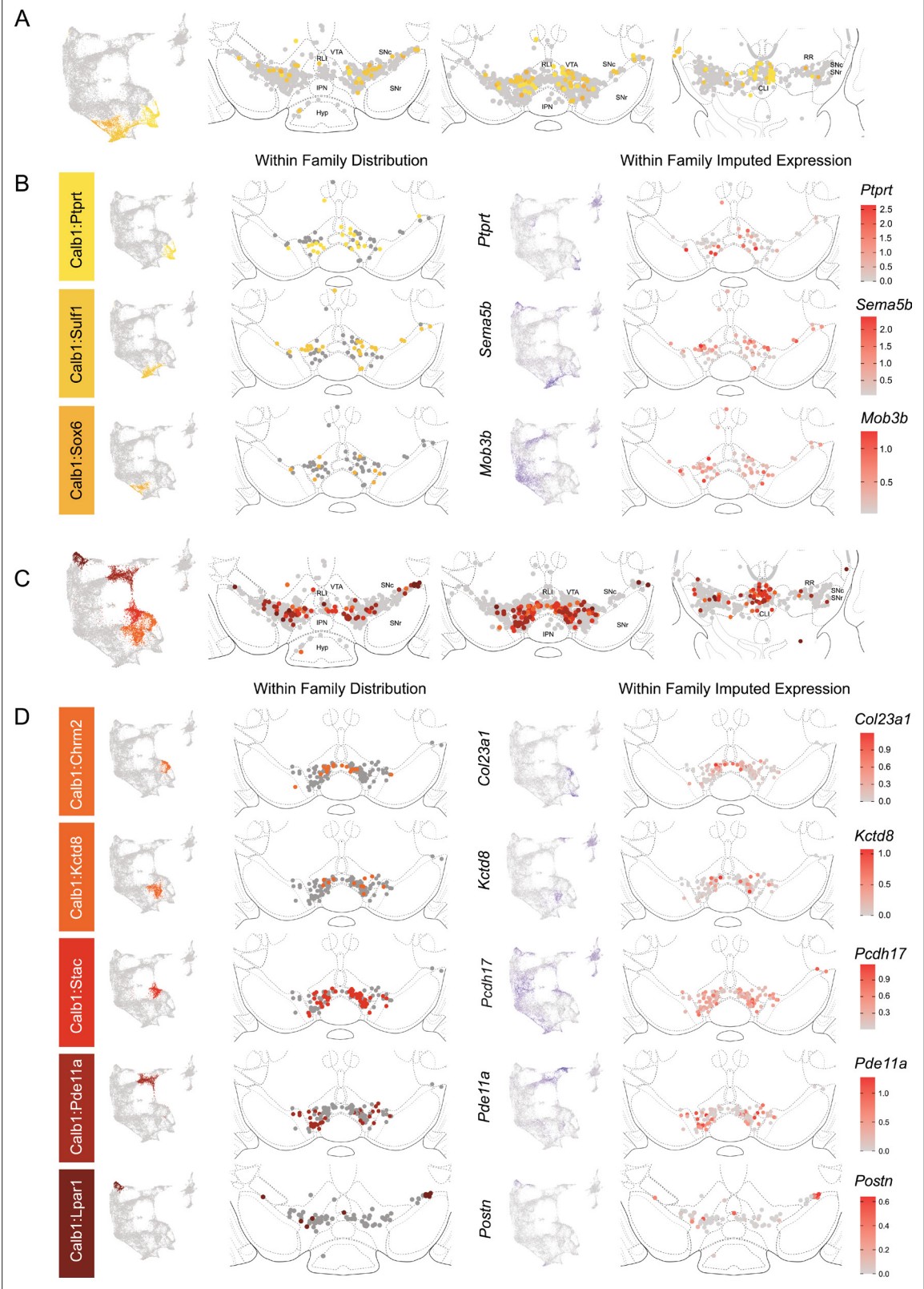

**Figure 6.** Spatial localization of the Calb1 family dopamine (DA) neurons. (**A**) Calb1 family of neurons was divided into two distinct branches based on hierarchical clustering. Distribution of Branch 1 of Calb1 family DA neurons along the rostral-caudal axis with all other DA cells colored in gray. (**B**) Location of individual clusters within Branch 1 of the Calb1 family. Only cells in the Branch 1 subset are shown. Relative cellular expression of genetic markers associated with each subtype is shown in a UMAP and brain section. (**C**) and (**D**) equivalent plots as (**A-B**), but for Branch 2 of the Calb1 family of DA neurons. Relative cellular expression of genetic markers associated with each subfamily is shown in a UMAP and a midbrain section.

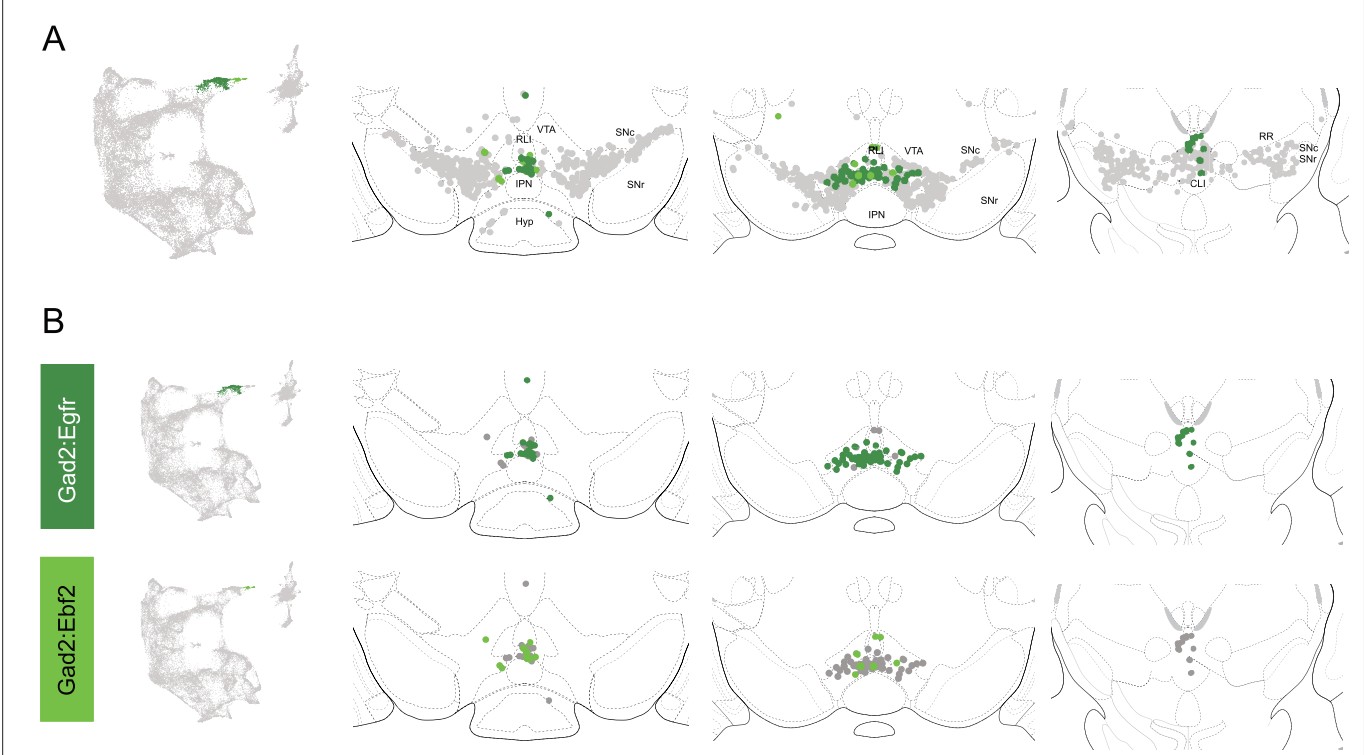

**Figure 7.** Spatial location of Gad2 family dopamine (DA) neurons. (**A**) Distribution of Gad2 family DA neurons along the rostral-caudal axis with other DA cells colored in gray. (**B**) Location of individual subtypes within the Gad2 family with only cells of the Gad2 family shown in gray.

possibility that these neurons co-release GABA. Further, they also expressed *Slc17a6*, revealing potentially multilingual DA neuron subtypes.

## LRRK2[G2019S] KI results in pan-dopaminergic gene expression changes consistent with deficits observed in this model, without altering DA subtype proportions

With cluster identities and locations defined, we next sought to explore the downstream gene expression characteristics of DA neurons in Lrrk2 mutants. The role of Lrrk2 in PD pathophysiology remains unclear (*Singh et al., 2019*), and Lrrk2 mutations are often proposed as being an indirect source of dysfunction for the DA system, such as through its high expression in glial or striatal cells (*Chen et al., 2020*; *Choi et al., 2015*; *Parisiadou et al., 2014*; *Cook et al., 2017*; *West et al., 2014*). However, previous studies have shown alterations in DA signaling or gene expression even when the mutant LRRK2 was expressed specifically in DA neurons (*Pallos et al., 2021*; *Liu et al., 2015*). Thus, as a first step in demonstrating the possibility that mutant Lrrk2 in DA neurons themselves can contribute to dysfunction, we mapped the expression of Lrrk2 across all DA neuron subtypes (*Figure 8A*). We found detectable expression across most subtypes, with the notable exception of Gad2[Egfr] neurons.

Given the distinctive properties of different DA neuron subtypes, we next sought to test if phenotypic changes in Lrrk2 mutant mice are driven by changes in the relative proportions of said subtypes. While previous studies have shown no overt DA neuron loss, it is possible that there was a selective reduction of a subtype(s) that would be undetectable by previous methods. We first graphed the proportions of each subtype within our control and mutant datasets (*Figure 8B*), and found remarkable similarity across these samples, consistent with previous reports showing no DA neuron loss in these mice (*Yue et al., 2015*). To further assess for changes in cell state among clusters across conditions, we next compared the similarity of each cluster to all other clusters across conditions (*Figure 8C*). MetaNeighbor analysis revealed that the most similar population to each wildtype cluster was its corresponding subtype in the Lrrk2 mutant samples. This suggests that the general subtype

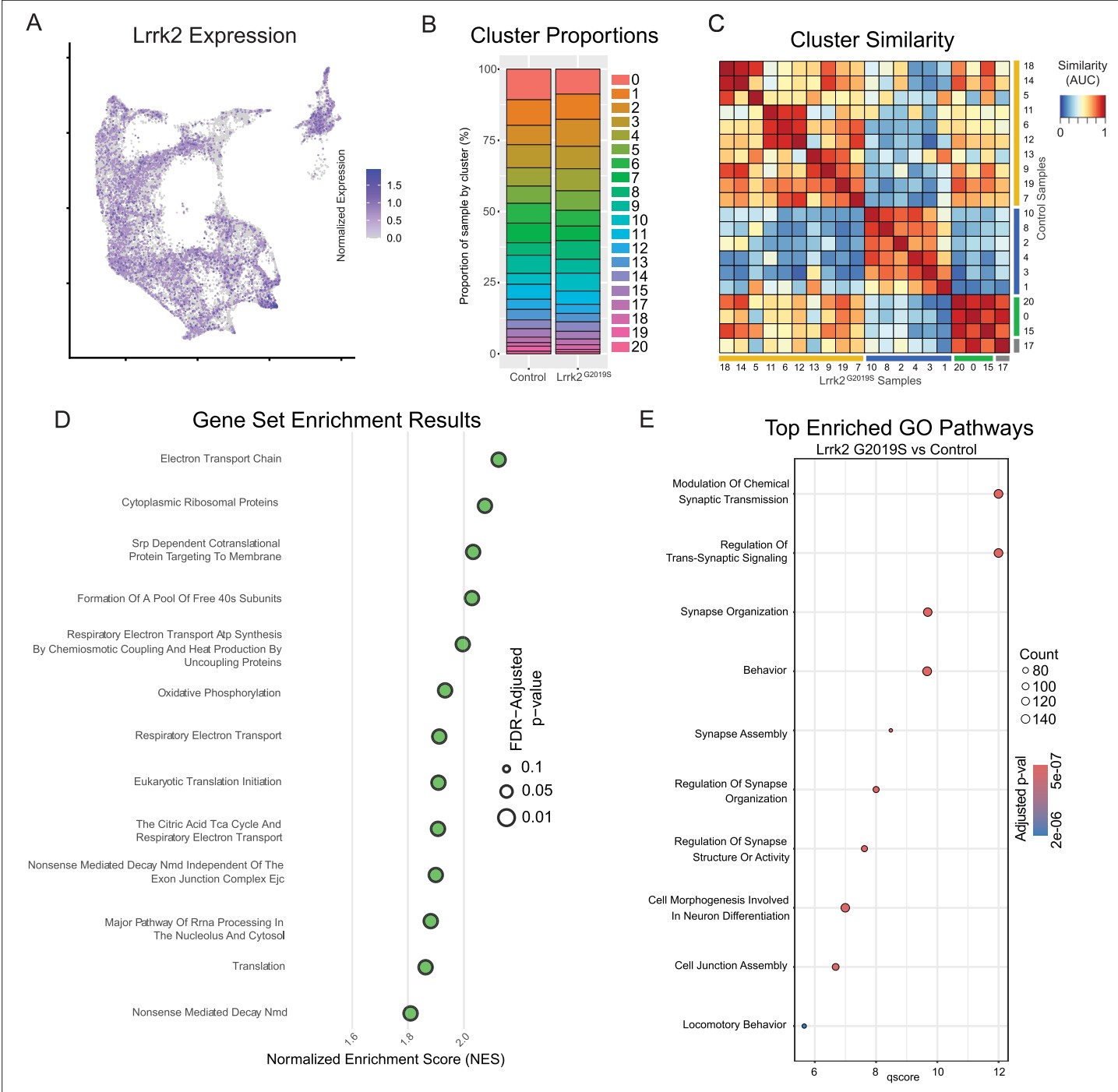

**Figure 8.** Global effects of Lrrk2$^{G2019S}$ mutation are observed across all clusters. (**A**) Expression pattern of Lrrk2 RNA, which is present in all clusters, though notably low in Gad2$^{Egfr}$ (cluster 15). (**B**) Cluster proportions in control and mutant samples. Relative proportions of each subtype within the dataset are relatively unchanged. (**C**) MetaNeighbor-generated cluster similarity heatmap shows that each cluster from control samples is more similar to its corresponding cluster in the mutant samples, suggesting there is no large-scale change in cell type across samples. (**D**) Gene set enrichment analysis (GSEA) results comparing all clusters across conditions. Several top enriched pathways in Lrrk2 mutants are related to mitochondrial energy production/oxidative phosphorylation. (**E**) Gene ontology (GO) results comparing all clusters across conditions. Top enriched pathways are related to synapse organization and function, consistent with previously described synaptic dysfunction in Lrrk2$^{G2019S}$ mutant mice.

The online version of this article includes the following figure supplement(s) for figure 8:

**Figure supplement 1.** Analysis of differentially expressed genes and their synaptic localizations and functional roles using SynGO.

organization is largely impervious to LRRK2$^{G2019S}$ perturbation, and that our clustering scheme is likely more reflective of cell type rather than cell state.

To assess pan-dopaminergic gene expression changes as a function of Lrrk2 genotype, we next looked at globally differentially expressed genes across our mutant and control samples. Calculating differentially expressed genes across all clusters revealed a large number of significantly enriched or diminished genes as a function of Lrrk2 mutation (1646 genes in total with BH corrected p-value <0.05, Wilcoxon rank sum test). The results of this differential expression analysis are plotted in *Figure 8—figure supplement 1A*, with full results and statistics available in *Supplementary file 2*. Notably, most of these changes appear to be small in magnitude; among FDR-significant genes, only 51 showed increased expression and seven showed decreased expression with log2 fold changes greater than 0.5 (*Figure 8—figure supplement 1A*). Two genes with high enrichment in Lrrk2 mutants, Mir124a-1hg and AC149090.1 were both of particular interest due to their potential relevance to PD pathogenesis. AC149090.1 is orthologous to the human PISD gene, which is involved in autophagy and implicated in mitochondrial dysfunction (*Buckley et al., 2023*; *Zhao et al., 2019*; *Thomas et al., 2018*). This gene has been proposed as a sensitive marker of biological aging in the brain (*Buckley et al., 2023*; *Jin et al., 2023*), an interesting finding given that age is the single largest risk factor for PD. Also of note, Mir124a-1hg is a host gene (though not the exclusive gene) for miR-124, a microRNA with purported neuroprotective effects that has been found to be downregulated in PD patients or some PD mouse models (*Zhang et al., 2022b*; *Yao et al., 2018*; *Han et al., 2019*; *Angelopoulou et al., 2019*). While the opposite effect was seen here (i.e. an upregulation of this gene in our model), one might speculate that miR-124 is initially increased during toxic metabolic insults as a compensatory response, given its proposed neuroprotective effects. In our prodromal model without observable degeneration, this could thus represent an early sign of cell stress. Conversely, in PD patients or overtly degenerative models, a lack of compensatory miR-124 or fulminant cell death among vulnerable cells could result in an observed decrease in miR-124 expression.

We next performed pathway enrichment analyses using Gene Set Enrichment Analysis (GSEA) and gene ontology (GO) to compare our Lrrk2 mutant samples to our controls. Using GSEA, we found 18 gene sets with significant differential expression (FDR-adjusted p-values <0.05). The top 13 of these enriched pathways (those with NES scores >1.8) are shown in *Figure 8D*. While a broad range of biological domains are found among these results, many of the top enriched pathways were related to energy production via mitochondrial oxidative respiration, remarkably similar to prior comparisons between vulnerable and resistant DA neuron populations (*Pereira Luppi et al., 2021*). Mitochondrial energy production pathways, and specifically the extraordinary bioenergetic burdens of DA neurons, have been heavily implicated in PD vulnerability (*González-Rodríguez et al., 2021*; *Surmeier et al., 2017*). Thus, the enrichment of these pathways in Lrrk2 mutants is particularly intriguing as it may reflect that the structural and functional alterations reported with LRRK2 mutations in patients (*Mortiboys et al., 2010*) and preclinical models (*Singh et al., 2021*; *Liu et al., 2021*) further predispose these cells to dysfunction and ultimately degeneration. Utilizing GO, we again found many pathways displaying significant differential expression between Lrrk2 mutants and controls. Among these, the top pathways are all associated with synaptic organization and functions (*Figure 8E*). This was particularly intriguing given that pre-synaptic dysfunction in Lrrk2 mutants such as endocytosis (*Nguyen and Krainc, 2018*; *Soukup et al., 2016*; *Islam et al., 2016*) or axonal cargo trafficking impairments (*Boecker et al., 2021*) could underpin the deficits in DA signaling observed in these mice (*Skelton et al., 2022*; *Pischedda and Piccoli, 2021*). Of note, decreased endocytosis has been observed in DA neurons but not cortical or hippocampal LRRK2$^{G2019S}$ primary neurons (*Pan et al., 2017*). These results, although in vitro, suggest cell type-specific synaptic effects on DA neurons, consistent with their vulnerability in LRRK2-related PD.

## Comparing individual subtypes across conditions allows insights into subtype-specific dysfunction

Given that global gene expression differences across our conditions appear to hold relevance for putative mechanisms of PD pathogenesis, we next sought to address the question of whether gene expression within any given subtype is specifically altered in our PD model. Although Lrrk2 mRNA seemed uniform across subtypes, downstream effects could be subtype-specific secondary to differential Lrrk2 kinase activity, distinct kinase targets, or superimposed differences in other properties of

each subtype that culminate in distinct downstream transcriptional alterations. Given the proposed vulnerability of Sox6 + DA neurons and relative resilience of Calb1 + neurons in PD (*Gaertner et al., 2022*; *Kamath et al., 2022*; *Pereira Luppi et al., 2021*), we first compared these two cluster families across Lrrk2 mutants versus control mice to see if any changes in DA neurons are specific to vulnerable or resistant cell types. We found 729 DEGs among the Sox6 family, and 679 among the Calb1 family (BH corrected p-value <0.05). Among those genes, we found 327 DEGs were shared between the Sox6 and Calb1 families (*Figure 8—figure supplement 1B*). The full list of DEGs and statistics is available in *Supplementary file 2*. Notably, AC149090.1 was highly significant in both cohorts, implying that the pan-DA differential expression of this gene was not driven by a particular subset of DA neurons. To further compare changes within cluster families, we once again performed GSEA and GO but on groups of clusters in isolation (*Figure 9A–B*). In line with our previous results, GO revealed pathways that largely corresponded to synaptic and axonal function, but with higher significance of results in the Sox6 family than in Calb1 or Gad2 families (*Figure 9A*). Within the Sox6 family, GSEA analysis showed an upregulation of energy production and metabolism pathways similar to those observed at the global level, as well as a downregulation of intercellular communication pathways (*Figure 9B*, top five upregulated and downregulated pathways shown for each family). Finally, to extend the granularity of our analyses, we applied the same GO pipeline to two individual clusters, Sox6[Tafa1] and Calb1[Stac], which have the highest expression of *Anxa1* in the SNc and VTA, respectively (*Figure 9—figure supplement 1A*). The Sox6[Tafa1] SNc subtype was of particular interest to us given recent results showing that *Anxa1* + ventral tier SNc neuronal activity is selectively correlated with acceleration in mice running on a treadmill, leading to the hypothesis that degeneration of these neurons may contribute to motor deficits seen in PD (*Azcorra et al., 2023*). The Calb1[Stac] VTA subtype was chosen as a comparator due to its similar expression profile, including key markers *Anxa1* and *Aldh1a1*, but within the relatively resistant VTA. Similar pathways were observed in each subtype, including several pathways once again corresponding to synaptic function, however with substantially higher significance values observed in the Sox6[Tafa1] cluster. Of note, while this enrichment is intriguing, the associations to such pathways cannot be interpreted as unique to this population, as technical limitations confound direct comparisons of DEGs among individual clusters due to variable cluster sizes and internal heterogeneity (see Methods for more details).

Next, we compared the synaptic compartment and biological functions of DEGs (BH adjusted p-values <0.05) in Lrrk2[G2019S] mutants versus controls using SynGO, a database for systematic annotation of synaptic genes (*Koopmans et al., 2019*). We focused primarily on the Sox6 and Calb1 families, but also on cluster Sox6[Tafa1] given that this population (SNc neurons with highest expression of *Anxa1*) is specifically linked to movement (*Azcorra et al., 2023*). Of the DEGs in the Sox6 family, Calb1 family, and Sox6[Tafa1] cluster, 24.82%, 20.19%, and 29.41% of these genes had synaptic localizations, respectively, with more significant synaptic associations among populations localized to ventral tier SNc (*Figure 9C*, *Figure 8—figure supplement 1C*). Further compartment categorization revealed that among these synaptic DEGs annotated by SynGO, 45.83% of Sox6 family (11.38% of total DEGs), 64.0% of Sox6[Tafa1] (18.82% of total DEGs) and 48.82% of Calb1 family (9.86% of total DEGs) DEGs have presynaptic localizations, raising the possibility of enhanced disruption of presynaptic function in acceleration-correlated DA neurons. From these presynaptic DEGs, 20/77 in the Sox6 family, 5/16 in Sox6[Tafa1], and 16/62 in the Calb1 family have specific annotations for active zones, the primary sites of evoked DA release. Annotations based on the biological functions showed that 20.83%, 28.0%, and 15.74% of Sox6, Sox6[Tafa1], and Calb1 DEGs, respectively, are associated with processes in the presynapse (*Figure 8—figure supplement 1C*). Overall, greater proportions of DEGs are associated with presynaptic locations in cells from vulnerable DA neurons (Sox6 family, and in particular, Sox6[Tafa1]), compared to less vulnerable ones (Calb1 family).

Given the differential effects of Lrrk2 mutation across subtypes, we also sought to map the association of gene expression in our dataset with co-expression of PD-associated risk loci identified by GWAS (*Nalls et al., 2019*) using a polygenic enrichment score method, scDRS (*Zhang et al., 2022a*). We first calculated a PD GWAS risk score for each individual cell, which revealed clear bias for higher risk scores among clusters that localize to the ventral SNc (*Figure 9F*). We next sought to extend the cell-level calculations of scDRS to explore the relative PD risk among different subtypes. To do so, we calculated the mean PD GWAS risk score for each cluster and utilized bootstrapping to generate a corresponding 95% confidence intervals for these mean values, allowing us to determine if the

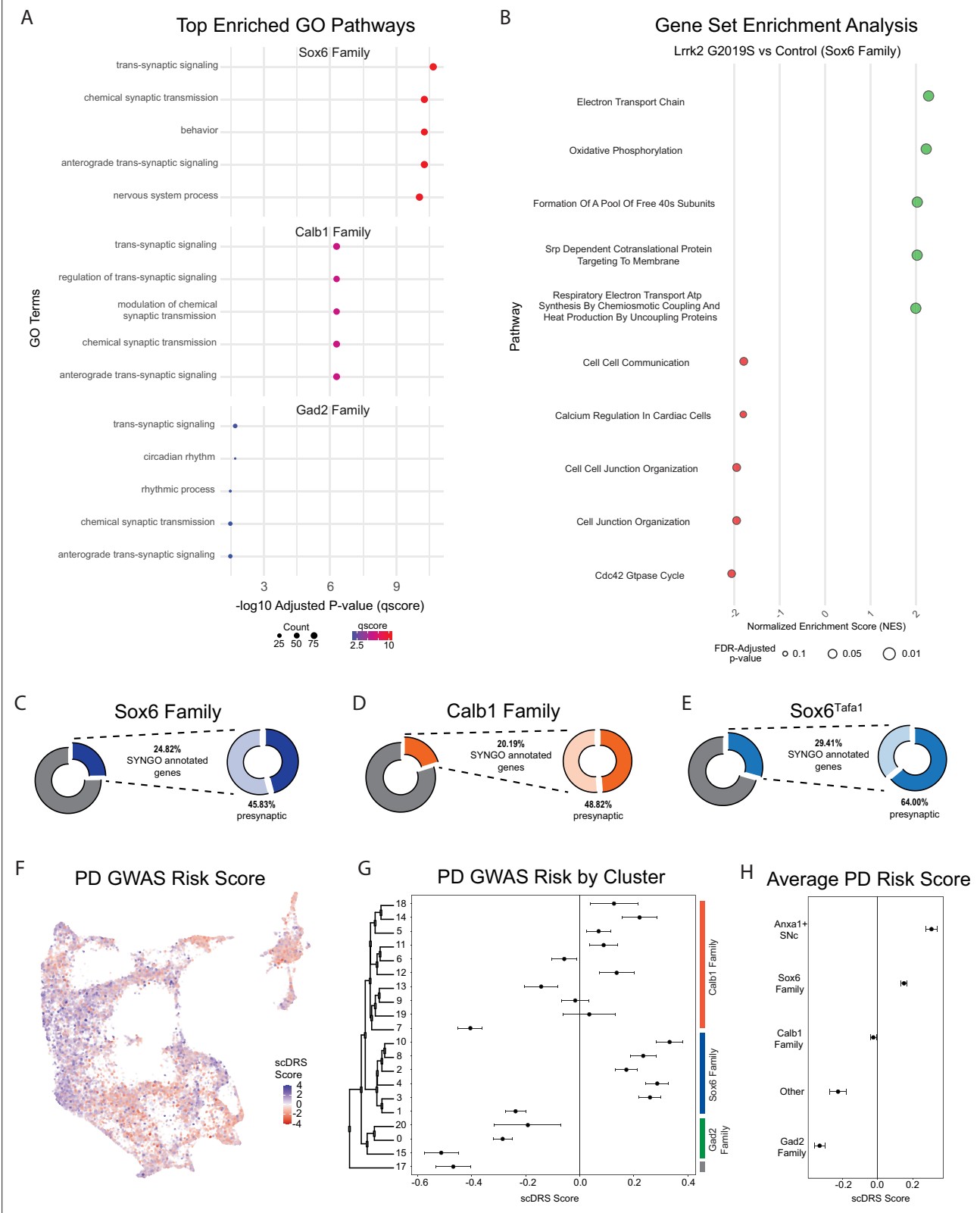

**Figure 9.** Changes in gene expression are observed in Parkinson's disease (PD)-implicated dopamine (DA) neuron subtypes in Lrrk2^G2019S mice. (**A**) Gene ontology (GO) results comparing each cluster family across conditions (top five enriched pathways shown). Several pathways are highly relevant to locomotor behavior and previously described dysfunction (such as synaptic organization pathways) in Lrrk2^G2019S mutant mice. (**B**) Gene Set Enrichment Analysis (GSEA) results comparing Sox6 cluster family across conditions. While twenty pathways were positively enriched in mutant Sox6 family clusters

*Figure 9 continued on next page*

Figure 9 continued

(BH-corrected p-values <0.05), several additional pathways were enriched in either direction using a less conservative cutoff of p<0.1. The top five positively and negatively enriched pathways with this less conservative cutoff are shown. Pathway enrichment was similar to global changes, but more pronounced in the Sox6 family of clusters. (C–E) Results of SynGO analyses for Sox6 family (9 C), Calb1 family (9D), or Sox6^Tafa1 clusters (9E). SynGO-annotated (i.e. synaptic) genes were enriched among differentially expressed genes (DEGs) for ventral substantia nigra (SNc) populations, particularly Sox6^Tafa1. Of note, among these synaptic genes, the Sox6^Tafa1 cluster showed much heavier enrichment for those localized to the presynapse, which may suggest subtype-specific presynaptic function in acceleration-corrected subtypes. (F) Feature plot of each cell's association with PD risk loci proposed by GWAS as calculated by scDRS scores. Clear differences are observed across clusters. (G) Mean and 95% confidence intervals for PD GWAS risk (scDRS scores) for each cluster. The Sox6 family of clusters is particularly elevated. (H) Average risk scores with 95% confidence intervals among cluster families, as well as the combination of clusters Sox6^Tafa1 and Sox6^Vcan, the two *Anxa1*-expressing SNc clusters, which showed the highest average risk scores.

The online version of this article includes the following figure supplement(s) for figure 9:

**Figure supplement 1.** Comparison of pathway analysis results between Anxa1 + clusters.

observed means were significantly different from the null. We confirmed significant risk associations in several clusters, particularly those in the Sox6 family (***Figure 9G***). Interestingly, the Gad2 family of clusters all showed significant negative associations with PD risk loci (***Figure 9G***). Of note, the highest scDRS scores were in clusters Sox6^Tafa1 and Sox6^Vcan, the only two SNc clusters that express *Anxa1*. By calculating the average scores for each cluster family as a whole, as well as these two Anxa1-expressing SNc clusters, we found that only the Sox6 family (and Anxa1-expressing SNc clusters within it) show significant associations with PD risk loci at a population level (***Figure 9H***).

Finally, due to the large number of potential comparisons to be made across clusters, cluster families, and genotypes, we have also developed an online tool that allows for exploration of our datasets, which we have termed Dopabase (URL: dopabase.org). Utilizing this tool, users have the ability to access many analysis and plotting functions to continue to explore populations of interest.

## Discussion

Our work provides five key advances. 1. We provide a large snRNA-seq dataset that provides high granularity to our taxonomic schemes. 2. We provide a user-friendly portal to query this dataset. 3. We plot the spatial location of most identified DA neuron subtypes, showing heterogeneity even within sub-domains in traditional anatomical areas. 4. We demonstrate that cluster proportions do not change in LRRK2^G2019S mice, although gene expression changes are observed, most notably in mitochondrial bioenergetics and synapse organization pathways in vulnerable DA subtypes. 5. PD GWAS risk is most prominent in Anxa1 SNc neurons.

We demonstrate the presence of 20 molecularly distinct clusters of midbrain DA neurons. These are hierarchically organized, the dendrogram being split into three levels, which we refer to as family, branch, and subtype. Family (Level 1) distinctions are driven by the expression of *Gad2/Kctd16* vs *Slc6a3/Ddc* high expressing cells. *Slc6a3/Ddc* high cells are further divided by *Sox6/Col25a1* and *Calb1/Ndst3* (Level 2) among other genes. It is likely that these fundamental divisions are, at least in part, set early in development in the mesodiencephalic floor plate, where *Sox6* demarcates progenitors with a substantially higher probability of ventral SNc and lateral VTA fate, and conversely, *Sox6*-progenitors have a higher probability of a *Calb1*+ fate (***Pereira Luppi et al., 2021***; ***Panman et al., 2014***). The Sox6 family is split into two branches with six subtypes, the Calb1 family is split into two branches with ten subtypes, whereas the Gad2 family can be further split into two subtypes. This scheme represents a more granular taxonomy than previous studies (***La Manno et al., 2016***; ***Kramer et al., 2018***; ***Hook et al., 2018***; ***Saunders et al., 2018***; ***Azcorra et al., 2023***; ***Poulin et al., 2014***; ***Kamath et al., 2022***), and bears some resemblance to a recent study (***Yaghmaeian Salmani et al., 2024***). Key congruencies between these studies include: (1) parsing midbrain DA system into broad groups of subtypes based on expression of *Sox6* and *Calb1*, and to a lesser extent *Gad2*, populations, (2) greater diversity observed in *Calb1* + clusters, (3) the identification of some consensus and homogenous subtypes. Key discrepancies include the extent to which each cluster is further subdivided and the final number and molecular signature of DA neuron subtypes. In the latter study (***Yaghmaeian Salmani et al., 2024***), they refer to levels as territories and neighborhoods. For simplicity of communication, we refer to the lowest level of our dendrogram of molecularly defined DA neuron clusters, as subtypes. However, we acknowledge that because of the common developmental origin from the

floor plate (*Blaess et al., 2011*; *Joksimovic et al., 2009*; *Yan et al., 2011*; *Andersson et al., 2006*), their high relatedness, the blurriness of boundaries between clusters, and the polythetic nature of clustering, referring to these groups as neighborhoods is an alternative and reasonable possibility. Notwithstanding these considerations, defining markers of some of these clusters and generating Cre/ Flp lines, may be useful to at very least enrich for DA neuron populations with distinctive properties as has recently been shown for the *Anxa1* + SNc neurons, which at a population level are selectively correlated with acceleration, while reward responses are almost entirely absent (*Azcorra et al., 2023*).

Our spatial MERFISH analysis provides the most granular representation of DA subtype locations to date. This resolution was obtained by the integration of snRNA-seq and MERFISH datasets. Our data complement current spatial transcriptomic datasets (*Zhang et al., 2023*; *Yao et al., 2023*; *Langlieb et al., 2023*). For instance, using MERFISH, *Yao et al., 2023* identified 10 clusters in the VTA (A10), 9 clusters in the SNc (A9), and 2 clusters in the RR although little detail on the molecular identity or location of these subtypes is provided. In addition, (*Langlieb et al., 2023*), provide a molecular atlas of the adult mouse brain in which they identified 13 clusters of midbrain DA neurons and located these using Slide-Seq. By contrast, *Yaghmaeian Salmani et al., 2024* used established markers to locate seven DA neuron territories and 16 neighborhoods. They report a distribution of *Gad2* + cells in the midline including CLi, RLi, and IF, which bears similarity with our Gad2 family. They also *Pdia5+/ Calb1+* DA neurons in the pars lateralis, likely representing an overlapping population with our Calb-1$^{Postn}$ and Calb1$^{Sulf1}$. Within the SNc, our data fits also well with the location of DA neurons that could be labeled with Anxa1-Cre, although here we show that in our Seurat-based clustering, Anxa1 expression is detected in Sox6$^{Tafa1}$ and to a lesser extent in Sox6$^{Vcan}$ cells. Taken together, with our increasing appreciation of DA neuron heterogeneity, a consensus is bound to arise on both the identity and location of DA neuron subtypes. Integration of all these emerging datasets will be necessary to achieve a comprehensive DA neuron taxonomy that will ultimately encompass all morphological, connectional, physiological, molecular, and neuroanatomical parameters.

A key finding of this study is the heterogeneity observed even within sub-domains of the traditional anatomical clusters. An example of this is the lateral SNc/SNpl. Current models depict the SN heterogeneity as a mediolateral gradient (*Cox and Witten, 2019*). Our data substantially refines this by showing that even within the lateral SNc/SNpl region, there are several distinct subtypes. One interesting finding is that two of the lateral subtypes from distinct families (Sox6 and Calb1), show some common gene expression signatures. For example, *Asic2* or *Ankfn1* are highly expressed in Calb1$^{Lpar1}$ and Sox6$^{Arhgap28}$ subtype, both of which are laterally located. One interpretation of this is that secondary gene expression programs may have been superimposed upon the early developmental sub-divisions, for instance during circuit assembly. A second example of heterogeneity within sub-domains is in the dorsolateral VTA. Again, we find enormous heterogeneity in this region, which is unaccounted for in previous models. It is likely that neurons in this region will have vastly distinct anatomical and functional properties, as exemplified by the complex intermixing of neurons with diverging axonal projections (*Lammel et al., 2008*). Careful intersectional genetic interrogation methods will need to be developed to distinguish these populations.

A strength of our study is that it utilizes advantages of each transcriptomic approach, the deep molecular profiling of individual cells using snRNA-seq and the spatial resolution of MERFISH. For instance, we relied on gene expression imputation to ascribe expression level to genes not covered/ detected in our MERFISH probe panel. Gene imputation as described by *Stuart et al., 2019* has been used in several recent studies integrating spatial and transcriptomic data (*Zhang et al., 2023*; *Yao et al., 2023*). It relies on identifying anchors that enable projection of MERFISH data onto the UMAP space of a snRNA-seq dataset and then uses neighboring cells to extrapolate the expression of genes not included in our probe panel. This approach was used to impute Sox6 expression, which accurately reflects what has been reported in prior immunofluorescence and in situ hybridization studies (*Avvisati et al., 2024*; *Poulin et al., 2018*; *Poulin et al., 2014*; *Pereira Luppi et al., 2021*; *Panman et al., 2014*). Moreover, imputed gene expression levels correlated strongly with MERFISH detected transcript for most genes further supporting our approach (*Figure 4—figure supplement 1* and *Figure 4—figure supplement 2*). Nevertheless, dataset integration has limitations that should be considered. First, imputed gene expression relies on the ability to identify reliable anchors linking the snRNA-seq and MERFISH datasets. These anchors are determined in part by the choice of genes included on probe panels and thus could indirectly influence the reliability of imputed gene

expression. Second, gene counts per cell in MERFISH are determined via segmentation of images, which is susceptible to artifacts and bias from centrally versus peripherally localized gene transcripts. In summary, although limitations are present in multi-modal transcriptomic analyses, merging these two approaches provided a molecular and spatial map of the DA system that could not have been resolved by either method alone.

We also attempted to address whether our clustering schemes are more reflective of cell types or cell states. We show that the proportions of subtypes are largely consistent between controls and mutants, and that mutant subtypes are most closely related to their control counterpart. This suggests that our taxonomic scheme is agnostic to a mild perturbation such as LRRK2[G2019S], suggesting that our clusters are reflective of cell types, rather than cell states. It is possible that with more severe perturbations, such as a toxin lesion, more substantial alterations of taxonomic schemes are observed (*Yaghmaeian Salmani et al., 2024*; *Tang et al., 2023*). However, we expect that for mild insults, day-to-day behavioral changes, or pharmacological paradigms, our clusters will be resistant to changes, although individual gene levels may vary. Nonetheless, we cannot definitively confirm that a given DA neuron cannot convert from one subtype to another. Ultimately, alternative approaches such as detailed fate mapping of clusters or RNAseq-based trajectory analyses with greater numbers of sampled cells could be used to resolve this question.

Our work describes a rich heterogeneity of DA neurons in the murine SNc that might serve as a reference for understanding patterns of degeneration in PD. The majority of SNc neurons belong to the Sox6 family, but some are from the Calb1 family. Several studies on post-mortem human PD brain have demonstrated a relative resilience of Calb1 + DA neurons (*Pereira Luppi et al., 2021*; *Yamada et al., 1990*; *Damier et al., 1999*). Calb1 + neurons in mice SNc and SNpl project most densely to the dorsomedial striatum (DMS) and tail of striatum (TS), respectively (*Gaertner et al., 2022*; *Poulin et al., 2018*). This could match the patterns of axon preservation observed in post-mortem PD brain, which show relative sparing of medial, ventral, and caudal regions of the caudate-putamen (*Kish et al., 1988*; *Kordower et al., 2013*). Indeed it is plausible that patterns of degeneration could be associated with distinct clinical presentations of PD – a recent study suggests that patients with relatively spared caudate projections have a higher probability of tremor (*Mendonça et al., 2024*). Additionally, we postulate that spared DA neurons projections may be a cellular substrate mediating the side-effects of levodopa particularly at high doses. In line with this, high DA release in the TS is associated with hallucination-like behaviors in mice (*Schmack et al., 2021*).

Given the relevance of LRRK2 to genetic and idiopathic PD, we examined the effects of LRRK2 on DA neuron gene expression to start appreciating the molecular pathophysiology of relevant subpopulations in PD. We relied on a KI LRRK2 model, expressing the G2019S pathogenic mutation due to the highest prevalence of this mutation over other mutations. Although these mice do not exhibit DA neuron loss, they do have nigrostriatal synaptic alterations within a physiological range (*Xenias et al., 2022*). Our transcriptomic interrogation has three main takeaways. First, the *Lrrk2* gene is expressed in most DA neurons in mice; in humans *LRRK2* transcript is enriched in the SOX6 population (*Kamath et al., 2022*). While previous work has shown LRRK2 protein expression in midbrain DA neurons (*West et al., 2014*; *Mandemakers et al., 2012*), its relative expression across DA neuron subtypes in mice requires further investigation. Thus, transcriptomic alterations observed within DA neurons are potentially cell-autonomous effects, albeit indirect, since LRRK2 functions as a protein kinase with phosphoregulation primarily affected. Second, while individual genes showed only modest differences, pathway analysis showed a strong dysregulation of synaptic pathways. This is conceptually aligned with prior work describing that LRRK2 phosphorylates various key presynaptic targets (not necessarily in DA axons) associated with SV cycle (*Pischedda and Piccoli, 2021*). Impairments of these processes at the nigrostriatal terminals may contribute to the DA release deficits reported in LRRK2 mutants (*Xenias et al., 2022*; *Yue et al., 2015*; *Tozzi et al., 2018*). Third, alterations in pathways associated with oxidative phosphorylation and energy production are in line with the described LRRK2 impact on mitochondrial structure and mitophagy in the LRRK2[G2019S] KI mice. As DA transmission changes and mitochondrial abnormalities are the two main and consistent phenotypes with the KI mouse model, our transcriptomic findings indicate that functional changes are at least partly linked with gene expression changes.

Taken together, our work complements perfectly the recent taxonomy of mouse (*Zhang et al., 2023*; *Yao et al., 2023*; *Langlieb et al., 2023*; *Yaghmaeian Salmani et al., 2024*) and human brain

cell types (*Siletti et al., 2023*). Focusing our efforts on the DA system allowed us to reveal a comprehensive transcriptomic signature of midbrain DA neurons as well as ascribe with precision their spatial location. Moreover, we provide a web resource to explore this dataset through multiple angles. We also uncovered cell type-specific transcriptomic changes in a prodromal model of PD revealing the molecular consequences of higher LRRK2 activity. Disentangling the molecular diversity of DA neurons, now defining 20 transcriptomic subtypes, raises important questions. For instance, what are the functional implications of this diversity and are these transcriptomic differences reflected in other cellular features? Decades of experiments might be needed to answer such questions, however, by defining genetic anchor points unique to each subtype, we have provided an avenue to target these populations in the mouse.

## Methods
### snRNA-seq
#### Animals
All animals used in this study were maintained and cared for following protocols approved by the Northwestern Animal Care and Use Committee (IS00015492). Cre mouse lines were maintained heterozygous by breeding to wild-type C57BL/6 mice (RRID:IMSR_CRL:027), with the exception of DAT-IRES-Cre (RRID:IMSR_JAX:027178, referred to as DAT-Cre in *Figure 1*), CAG-Sun1/sfGFP (RRID:IMSR_JAX:021039, referred to as RC-LSL-Sun1/GFP in *Figure 1*), which was maintained homozygous and crossed to Lrrk2$^{G2019S}$ mice (RRID:IMSR_JAX:030961) only for generation of experimental mice for nuclei isolation. Both males and females were used in equal number for all RNAseq experiments.

#### Sample preparation
To isolate nuclei for snRNA-seq library generation, n=8 6-mo-old DAT-IRES-CRE (RRID:IMSR_JAX:027178), CAG-Sun1/sfGFP (RRID:IMSR_JAX:021039), Lrrk2$^{G2019S+/wt}$ or littermate Lrrk2$^{wt/wt}$ mice (2 female control, 2 male control, 2 female mutants, 2 male mutants) were sacrificed and rapidly decapitated for extraction of brain tissue as previously described for isolation of GFP+ (i.e. dopaminergic) nuclei. A total of four independent samples (2 control, 2 mutants; equal sexes per sample) were isolated using n=2 pooled mice in each. Each sample was processed in its own GEM well to produce four distinct libraries that were subsequently sequenced and analyzed as described below. A 2 mm thick block of ventral midbrain tissue was dissected out from each mouse and collected for isolation. Tissue was dounce homogenized in a nuclear extraction buffer 10 mM Tris, 146 mM NaCl, 1 mM CaCl$_2$, 21 mM MgCl$_2$, 0.1% NP-40, 40 u/mL Protector RNAse inhibitor (Roche 3335399001). Dounce homogenizer was washed with 4 mL of a washing buffer (10 mM Tris, 146 mM NaCl, 1 mM CaCl$_2$, 21 mM MgCl2, 0.01% BSA, 40 U/mL Protector RNAse inhibitor) and filtered through a 30 uM cell strainer. After three rounds of washing by centrifugation at 500 g for 5 min, nuclei pellets were resuspended resuspension buffer (10 mM Tris, 146 mM NaCl, 1 mM CaCl2, 21 mM MgCl$_2$, 2% BSA, 0.02% Tween-20) and filtered through a 20 uM strainer. This nuclei suspension was loaded onto a MACSQuant Tyto HS chip and diluted with 1 x PBS. Nuclei were sorted using gates set for isolation of GFP + singlet nuclei. Sorted nuclei were subsequently used for preparation of four 10 X Genomics Chromium libraries. Protocol can be found at dx.doi.org/10.17504/protocols.io.14egn6wryl5d/v1.

Library preparations were performed by the Northwestern University NUSeq Core Facility. Nuclei number and viability were first analyzed using Nexcelom Cellometer Auto2000 with AOPI fluorescent staining method. Sixteen thousand nuclei were loaded into the Chromium Controller (10 X Genomics, PN-120223) on a Chromium Next GEM Chip G (10 X Genomics, PN-1000120), and processed to generate single nucleus gel beads in the emulsion (GEM) according to the manufacturer's protocol. The cDNA and library were generated using the Chromium Next GEM Single Cell 3' Reagent Kits v3.1 (10X Genomics, PN-1000286) and Dual Index Kit TT Set A (10 X Genomics, PN-1000215) according to the manufacturer's manual with following modification: PCR cycle used for cDNA generation was 16 and the resulting PCR products was size-selected using 0.8 X SPRI beads instead of 0.6 X SPRI beads as stated in protocol. Quality control for constructed library was performed by Agilent Bioanalyzer High Sensitivity DNA kit (Agilent Technologies, 5067–4626) and Qubit DNA HS assay kit for qualitative and quantitative analysis, respectively.

The multiplexed libraries were pooled and sequenced on an Illumina Novaseq6000 sequencer with paired-end 50 kits using the following read length: 28 bp Read1 for cell barcode and UMI and 91 bp Read2 for transcript. Raw sequence reads were then demultiplexed and transcript reads were aligned to mm10 genome using CellRanger (v7.0.1 – RRID:SCR_017344 – https://www.10xgenomics.com/support/software/cell-ranger/latest). snRNA datasets can be found at GEO accession GSE271781.

## Data curation and analysis

Analysis was performed using R (RRID:SCR_001905). Outputs from CellRanger were read into Seurat (version 5.0.1; RRID:SCR_007322) using the Read10X command for each sample. Numbers of UMIs, features and ribosomal reads, and mitochondrial reads were plotted for each dataset and used to determine cutoffs for quality control pre-filtering of each sample. The following QC filtering commands were used in R: control1 <- subset(control1, subset = nFeature_RNA >1200 & nFeature_RNA <7800 & percent.mt <0.5 & nCount_RNA <29000 & percent.ribo <0.5); lrrk1 <- subset(lrrk1, subset = nFeature_RNA >1200 & nFeature_RNA <6500 & percent.mt <0.5 & nCount_RNA <26000 & percent.ribo <0.5); control2 <- subset(control2, subset = nFeature_RNA >800 & nFeature_RNA <5000 & percent.mt <0.5 & nCount_RNA <15000 & percent.ribo <0.5); lrrk2 <- subset(lrrk2, subset = nFeature_RNA >1000 & nFeature_RNA <7500 & percent.mt <0.5 & nCount_RNA <20000 & percent.ribo <0.5) (post-filtering violin plots shown in, *Figure 1—figure supplement 1*). The male and female datasets were then normalized and integrated using the SCTransform V2 method *Choudhary and Satija, 2022* in Seurat V5 (*Hao et al., 2024*). Cells were initially clustered using 35 principal components at a resolution of 0.8. Plotting clusters on UMAP showed one small cluster exceedingly distant from all other clusters, which was suspected to represent EW nucleus cells and was subsequently removed. Afterwards, the integration and clustering pipeline was repeated without these cells to remove their impact on PCA. Final clustering was achieved using the Find-Neighbors() command with the following parameters: dims = 1:32, reduction = 'cca,' n.trees=500, k.param=40, and with the FindClusters() command with the following parameters: resolution = 0.8, algorithm = 1, group.singletons=TRUE, graph.name = 'SCT_snn.' UMAP reduction was calculated using the RunUMAP command with the following parameters: reduction = 'cca,' dims = 1:32, reduction.name = 'umap.cca,' n.epochs=500, min.dist=0.2, n.neighbors=1000. In total, the integration resulted in a final dataset of 28532 nuclei, with a median UMI count of 7750.5 and median of 3056 features. All clusters were represented in both male and female samples (*Figure 1—figure supplement 1*).

Cluster dendrogram was produced using the Seurat BuildClusterTree() function with the same parameters used for clustering; dims = 1:32, reduction = 'cca.' Differential expression between branches of the dendrogram were calculated using FindMarkers() with the following parameters to maximize detection of DEGs with binary differences across branches, rather than genes expressed in both branches at different levels; min.diff.pct=0.25, logfc.threshold=1, min.pct=0.2. The top positively and negatively enriched genes were then plotted for manual curation of branch point defining markers that best distinguished each branch, with preference given to genes previously described in DA neuron subtype literature.

Sex of original mice was inferred for each cell by calculating the expression ratio of genes *Uty* and *Eif* relative to *Tsix* and *Xist* for each cell, revealing roughly equal proportions of male and female cells in the final dataset (*Figure 1—figure supplement 1*). All clusters were represented in each individual sample (*Figure 1—figure supplement 1*), and with no overt differences in proportions across conditions (*Figure 1—figure supplement 1*, *Figure 8B*). Clusters 21 and 16 were removed from downstream analyses due to high expression of glial markers Atp1a2 and Mbp, respectively (*Figure 1—figure supplement 1*). All differential expression calculations were also performed in Seurat using the FindMarkers() command; individual parameters for differential expression at each step of our analysis are provided in the Awatramani Lab Github page (https://github.com/AwatramaniLab/Gaertner_Oram_etal copy archived at *AwatramaniLab, 2025*). Of note, the default Seurat differential expression calculation method is known to overestimate significance of differential gene expression due to treating individual cells as independent samples. While pseudo bulk differential gene expression is preferable given these limitations, the number of independent 10 X library preparations (2 per group) does not provide sufficient sample numbers for the recommended minimums for pseudo bulk differential gene expression approaches, and thus the default Seurat methodology was

used; caution should be used when interpreting p values of individual DEGs such as those shown in the volcano plot of (*Figure 8—figure supplement 1A*).

ShinyCell (RRID:SCR_022756) web browser was created using the SCT assay of the Seurat Object described above (*Ouyang et al., 2021*). The code can be found at the Awatramani Lab Github page.

Mapping data query sets were done using TranserData() seurat function. The dataset and clusters from Azcorra & Gaertner et al. (Dataset can be found in the Gene Expression Omnibus GSE222558) *Azcorra et al., 2023* was used as the reference, and the query dataset was the one described in *Figure 1*. Any cells clustered with a prediction.max.score <0.5 were removed from the data set (2.4% of cells removed). The exact workflow can be found at the Awatramani Lab Github page.

Sankey plots were made using the Network3D package in R and the sankeyNetwork function. The 'source' and 'target' nodes are the corresponding cluster labels. 'Value' is set as the number of cells clustered in both the source and target nodes. The exact workflow can be found at the Awatramani Lab Github page.

## Cluster stability and homogeneity analyses

To quantify how homogenous each cluster is and uncover potential further subdivisions that may exist within our clusters, we applied two approaches highly similar to those used for our previously described dataset (*Azcorra et al., 2023*). First, we generated a measure of stability for each cluster through random downsampling and reclustering of the dataset using the same custom R scripts as previously described (*Azcorra et al., 2023*). Doing so provided cluster stability metrics (*Figure 2—figure supplement 1*) representing the propensity for cells within a cluster to continue to co-cluster when data is removed.

To understand potential sources of lower cluster stability values (e.g. two clusters being grouped together as one, or additional small clusters being divided into two adjacent clusters), we utilized the ClusTree (v0.5.1; RRID:SCR_016293 – https://github.com/lazappi/clustree; copy archived at *Zappia et al., 2023*) R package to plot clustering at ten incremental resolutions of 0.1 through 1.1 (*Figure 2—figure supplement 1*).

## Scanpy

Analysis done using Python (v 3.10.13 *Wolf et al., 2018* - RRID:SCR_008394). Cells were clustered using the scanpy toolkit (Version #1.10.1 RRID:SCR_018139; https://scanpy.readthedocs.io/en/stable/). Outputs from CellRanger were read into Scanpy using scanpy.read_10 x_mtx(). Cells were filtered with the same parameters as described above in the Seurat workflow. Datasets were individually clustered using the Leiden clustering algorithm. The four datasets were then integrated using scanpy.tl.ingest(). Each dataset was mapped onto Control 1. The exact workflow can be found at the Awatramani Lab Github page.

## Cluster similarity across control and Lrrk2 samples

To assess the molecular signature similarity between the clusters from Control and Lrrk2$^{G2019S}$ samples, we first corrected for dropouts the expression matrix of each sample using Adaptively-thresholded Low Rank Approximation (ALRA) (*Linderman et al., 2022*). Then, using MetaNeighbor standard pipeline (*Crow et al., 2018*), we picked the intersect of variable genes across all but the top decile of expression bins for Control and Lrrk2 samples. On the 5282 selected genes, MetaNeighbor was performed to assess an AUROC score of clusters similarity. The exact workflow can be found at the Awatramani Lab Github page.

## Pathway enrichment analyses

To evaluate pathways and gene sets enriched in subtypes or across conditions, we utilized the fgsea (v3.19; https://bioconductor.org/packages/release/bioc/html/fgsea.html; RRID:SCR_020938) and clusterProfiler (v4.8.3; RRID:SCR_016884 – https://doi.org/10.18129/B9.bioc.clusterProfiler) R packages. Pathways for GSEA analysis (mouse canonical pathways gene sets) were obtained from MSigDB (v2023.2.Hs, 2023.2 .Mm; https://www.gsea-msigdb.org/gsea/msigdb; RRID:SCR_016863). Custom R scripts were created for visualization of GSEA results and are available in the deposited code for this paper. fgsea was run using the following parameters: nperm = 1000, stats = ranks, minSize = 25, maxSize = 500. Differential expression calculations for fgsea package input were performed using

Seurat's FindMarkers command to generate a ranked list of genes, with filtering out of low-expression genes based on a minimum of 10% detection in either population being compared.

For gene ontology, the biological processes gene sets were obtained using the org.Mm.eg.db R package (v3.19; https://biocondutor.org/packages/release/data/annotation/html/org.Mm.eg.db.html; RRID:SCR_002774), and GO (part of clusterProfiler package) was run using default settings with BH adjustment of p-values and utilizing FDR-adjusted significant (p<0.05) DEGs for the respective comparisons. A custom genetic background was set for every individual comparison calculated to ensure hypergeometric testing was not simply enriched for cell type specific pathways; for comparisons of subsets within the dataset (i.e. a family or cluster) across conditions, a minimum detection level of 10% of cells was used to define the genetic background. These same thresholds were applied to filter the DEG lists used as input for GO. Results were visualized using the enrichplot R package (v3.19; https://www.bioconductor.org/packages/release/bioc/html/enrichplot.html; RRID:SCR_006442).

## scDRS cell-level scoring of GWAS risk associations

The scDRS score represents the relative association for each individual cell's expression profile (among all other cells in the dataset) with PD risk loci by utilizing the underlying SNPs and associations described in GWAS summary statistics. Gene expression data ('corrected' UMI counts) was extracted from the SCT assay. Expression matrices, cell metadata, and feature metadata were loaded using Pandas (v2.2.2; https://pandas.pydata.org/; RRID:SCR_018214) in Jupyter (Jupyter: v7.2.1; https://jupyter.org/install; RRID:SCR_018315) notebook. The data were converted into an AnnData object using the anndata package (AnnData v0.10.8; https://anndata.readthedocs.io/en/latest/; RRID:SCR_018209) and stored in HDF5 format. Normalization and scaling were performed using the Scanpy package (v1.8.1) (PMID: 29409532). Total counts for each cell were normalized to 10,000 reads using the sc.pp.normalize_total function and log-transformed using sc.pp.log1p. The sc.pp.highly_variable_genes function (parameters: min_mean = 0.0125, max_mean = 3, and min_disp = 0.5) was used to identify highly variable genes. Finally, the anndata object was scaled using the sc.pp.scale function, with a max_value parameter of 10.

GWAS summary statistics for Parkinson's Disease risk were obtained from EBI GWAS Catalog Study ID: GCST009325 (PMID: 31701892) and processed using the MAGMA tool on FUMA's online platform (v1.5.2; https://fuma.ctglab.nl/snp2gene; RRID:SCR_017521; RRID:SCR_005757) to generate a scored list (z-scores) of significant genes. Default settings were used with a 10×10 gene window. Scored gene list was processed using the munge-gs function of the scDRS package (v1.0.2; https://martinjzhang.github.io/scDRS/) to create.gs gene sets (PMID: 34431100) and loaded into the Jupyter Notebook using the scdrs.util.load_gs function. To align with the anndata object data, the gene set's human genes were mapped to mouse orthologs and intersected with genes present in the dataset. Preprocessing was completed using the scdrs.preprocess function, binning genes by mean and variance with parameters n_mean_bin = 20 and n_var_bin = 20. Scoring for Parkinson's disease was performed using the scdrs.score_cell function (parameters: ctrl_match_key = mean_var, n_ctrl = 1000, weight_opt = vs, return_ctrl_raw_score = False, and return_ctrl_norm_score = True).

Since scDRS does not include a native method for population-level p-values, we calculated a mean scDRS score for each cluster or family by averaging its component cells and assessed for significance by creating a bootstrapped confidence interval for these means. A score of 0 represents the null of no association between gene expression and PD risk loci, and thus if the 95% confidence interval does not overlap 0, the mean scDRS score for a given group can be regarded as significant as there is a less than 5% chance of the true group mean containing the null. Bootstrapped 95% confidence intervals for mean score within clusters or cluster families were calculated in R using 10,000 permutations for each group and plotted as mean scDRS score per group with CI represented as error bars.

Parkinson's Disease MAGMA Analysis Ranked Genes dataset can be found at 10.5281/zenodo.13076447.

## SynGO analyses

SynGO analysis protocol is described in detail at https://doi.org/10.17504/protocols.io.x54v92r94l3e/v1.

## MERFISH

### Animals

Experiments were approved by the Montreal Neurological Institute Animal Care Committee and conducted according to guidelines and regulations from the Canadian Council on Animal Care (Animal protocol number MNI-8132).

Adult C57Bl/6 male mice (RRID:IMSR_CRL:027) aged 2–3 mo were used for the MERFISH experiments. Animals were housed at the Montreal Neurological Institute (MNI) on a 12 hr-12 hr light-dark cycle (light cycle 7:00 to 19:00) with ad libitum access to food and water. Experiments were approved by the MNI Animal Care Committee and conducted according to guidelines and regulations from the Canadian Council on Animal Care.

### Gene selection and panel assembly for MERFISH

Hybridization probes were generated by Vizgen using their custom gene portal. DA neuronspecific genes were selected based on differential gene expression analysis from the 20 clusters identified in the single-nuclei sequencing data. Also included were: (1) general markers of glutamatergic, GABAergic, cholinergic, serotonergic neuron populations, (2) markers of non-neuronal cell types, (3) neuropeptide precursor genes and neuropeptide receptors, and (4) general ion channels. The list of all probes on the panel can be found in *Supplementary file 1*.

### Tissue preparation and tissue imaging for MERFISH

Mice were perfused with 1 x PBS followed by 4% paraformaldehyde (PFA). Brains were extracted and post-fixed in PFA for 16 hr, embedded in O.C.T. (Tissue-Tek O.C.T.; 25608–930, VWR), and stored at −80 °C until sectioning. Frozen brains were sectioned at −18 °C on a cryostat (Leica CM3050S). Within three animals we collected 10um-thick sequential coronal brain sections every 200 μm on specialized glass slides (approximately 5–6 sections per mouse). Seven sections were chosen with the best morphology across the midbrain for processing. Sample preparation was completed using Vizgen gene imaging kits provided according to the manufacturer. All steps were performed under RNAse-free conditions. Sections were first washed with 1 x PBS three times before incubation in 70% Ethanol overnight at 4 °C to permeabilize cell membranes. The following day each section was washed with sample wash buffer followed by an incubation in formamide wash buffer at 37°C for 30 min. The gene panel mix containing hybridization probes was then placed on top of the tissue and placed in a humidified 37°C chamber for 36–48 hr. Afterwards, the tissue was incubated with formamide buffer for 30 m at 47°C twice before washing with sample wash buffer. The hybridized tissue was then embedded in a 0.05% ammonium persulfate gel and cleared overnight in a solution of Proteinase K (NEB Cat# P8107S) at 37°C. The following morning the sample was incubated in a solution containing DAPI and PolyT stain to aid in cell segmentation. Once processed, brain sections were prepped for imaging using the commercial Merscope system developed by Vizgen. 500-gene panel cartridges were thawed in a 37 °C water bath for 1 hr before the start of imaging. Mouse RNAse inhibitor (NEB Cat# M0314) was added to imaging activation buffer and added to the cartridge fluidic system before loading into the Merscope to prime the fluidics chamber. Slides were removed from wash buffer and placed in the imaging capsule, connected to the fluidics chamber. Sections were imaged according to the manufacturer (Vizgen). The exact protocol for *tissue preparation and tissue imaging for MERFISH* can be found at dx.doi.org/10.17504/protocols.io.14egn6opyl5d/v1.

### Preprocessing and quality control of MERFISH data

Cell by gene matrices were processed in R using the Seurat V5 package (*Hao et al., 2024*). To exclude cells that underwent poor cell segmentation we removed cells that had >500 $\mu m^3$ and <4000 $\mu m^3$ volumes. To remove cells that show low transcript identity or falsely identified cells we removed any cell that had <40 total RNA transcripts detected. The remaining cells then underwent SCT normalization (*Choudhary and Satija, 2022*) before running RPCA integration to integrate datasets from individual sections. Shared nearest neighbor clustering was completed using the first 37 principle components using a resolution of k=0.7. Genetic markers of each cluster were determined using a roc test and was used to annotate our dataset. High-level annotations for glia and neuron types were confirmed using the SCType package in R (v3.0; https://sctype.app/). The putative DA cluster was

subsetted, re-normalized, and subclustered using the first 19 principle components and a resolution of k=0.65. Gene markers of each subcluster was determined using a ROC test and the top three genes from each cluster were identified based on the high positive log-fold change with a low p-values (adjusted p-value $<10^{-50}$). Code can be found on the Awatramani Lab Github page (https://github.com/AwatramaniLab/Gaertner_Oram_etal; copy archived at *AwatramaniLab, 2025*).

## Integration of MERFISH data with scRNA-seq data

To integrate MERFISH data with snRNA-seq data we utilized the weighted-nearest neighbour algorithm defined in Stuart, Butler, et al (*Stuart et al., 2019*). Briefly, anchor points were established between the two datasets using the FindTransferAnchors and MapQuery functions in Seurat. This allowed for a projection of the MERFISH data into the scRNA-seq space and a transfer of cluster identity between the two. Importantly, a label transfer prediction score was assigned to each MERFISH cell and cells with scores greater than 0.5 were classified as high confidence cells. Whole genome expression was imputed onto MERFISH cells using snRNA-seq data using the TransferData function in Seurat using the snRNA-seq normalized data as a reference. Cluster identifications were compared between transfer data and MERFISH subclustering using a Sankeyplot in R. Briefly, the number of cells that transferred between two pairs of clusters were calculated and expressed as a proportion within each MERFISH cluster. Only flows >10% were included in the visualization. Spatial visualizations of DA subtypes were generated using the ImageDimPlot function in Seurat and overlaid on the Allen Brain Atlas mouse anatomical template or the Paxinos mouse brain coronal axis using affinity designer. Gene expression values were obtained using the ImageFeaturePlot function in Seurat. Code can be found on the Awatramani Lab Github page (https://github.com/AwatramaniLab/Gaertner_Oram_etal; copy archived at *AwatramaniLab, 2025*).

### MERFISH anatomical localization

We utilized the Visualizer software (v2.2, Vizgen Inc https://portal.vizgen.com/resources/software) to delineate mouse midbrain regions into 11 distinct regions: the caudal linear nucleus (CLi), interfascicular nucleus (IF), interpeduncular nucleus (IPN), periaqueductal gray region (PAG), posterior hypothalamus (PH), rostral linear nucleus (RLi), retrorubral field (RR), substantia nigra pars compacta (SNc), Substantia nigra pars reticulata (SNr), Superior Colliculus (SC) and the ventral tegmental area (VTA). Brain regions were manually sectioned based on the intensity of DAPI imaging as well as the presence of *Th* transcripts compared to the Allen Brain Atlas. Cells within each ROI were counted to generate the localization of each family and cluster by counts and proportions.

MERFISH data can be found at DOI #: 10.5281/zenodo.12636327.

## Acknowledgements

ZG and CO performed snRNA-seq and MERFISH experiments and data analysis. CB, ES, AS, and CC performed data analysis and shinycell app construction. LP and DD provided scientific input on DA classification and LRRK2 alterations. ZG, CO, LP, JFP, RA wrote the manuscript. We would like to acknowledge Dr. Kevin Petrecca, Dr. Phuong Le, and Michael Luo for generously providing access to Merscope and technical advice. RA, DD, AS, and LP were funded by Aligning Science Across Parkinson's (ASAP-020600). For the purpose of open access, the authors have applied a CC-BY 4.0 public copyright license to this manuscript. RA was also funded by 1R01NS119690-01, P50 DA044121-01A1. LP was funded by R01 NS097901. ZG was funded by NINDS 1F31NS115524-01A1. JFP was funded by CIHR (PJT-183760), HBHL, Parkinson Canada. CB was funded by Parkinson Canada, FRQS, and CIHR.

# Additional information

## Funding

| Funder | Grant reference number | Author |
| --- | --- | --- |
| Aligning Science Across Parkinson's | ASAP020600 | Zachary Gaertner<br>Amanda Schneeweis<br>Chuyu Chen<br>Daniel Dombeck<br>Loukia Parisiadou<br>Rajeshwar Awatramani |
| National Institutes of Health | 1R01NS119690-01 | Rajeshwar Awatramani |
| National Institutes of Health | P50 DA044121-01A1 | Rajeshwar Awatramani |
| National Institutes of Health | R01 NS097901 | Loukia Parisiadou |
| National Institutes of Health | 1F31NS115524-01A1 | Zachary Gaertner |
| Canadian Institutes of Health Research | PJT-183760 | Jean-Francois Poulin |
| Healthy Brains Healthy Lives | | Jean-Francois Poulin |
| Parkinson Canada | | Cameron Oram<br>Jean-Francois Poulin |
| Fonds De La Recherche Scientifique - FNRS | | Cameron Oram |
| Canadian Institutes of Health Research | | Cameron Oram |

The funders had no role in study design, data collection and interpretation, or the decision to submit the work for publication.

## Author contributions

Zachary Gaertner, Cameron Oram, Conceptualization, Data curation, Software, Formal analysis, Supervision, Validation, Visualization, Methodology, Writing - original draft, Writing – review and editing; Amanda Schneeweis, Resources, Software, Visualization, Project administration; Elan Schonfeld, Data curation, Formal analysis; Cyril Bolduc, Chuyu Chen, Data curation, Formal analysis, Visualization; Daniel Dombeck, Loukia Parisiadou, Conceptualization, Supervision, Writing – review and editing; Jean-Francois Poulin, Rajeshwar Awatramani, Conceptualization, Supervision, Funding acquisition, Writing – review and editing

## Author ORCIDs

Zachary Gaertner ![ORCID] http://orcid.org/0000-0002-1760-6549
Amanda Schneeweis ![ORCID] https://orcid.org/0000-0003-4141-6064
Elan Schonfeld ![ORCID] https://orcid.org/0000-0001-7368-1562
Chuyu Chen ![ORCID] https://orcid.org/0000-0001-5666-5173
Daniel Dombeck ![ORCID] https://orcid.org/0000-0003-2576-5918
Loukia Parisiadou ![ORCID] https://orcid.org/0000-0002-2569-4200
Jean-Francois Poulin ![ORCID] http://orcid.org/0000-0002-1039-4985
Rajeshwar Awatramani ![ORCID] https://orcid.org/0000-0002-0713-2140

## Ethics

All animals used in this study were maintained and cared following protocols approved by the Northwestern Animal Care and Use Committee (IS00015492). Experiments were approved by the Montreal Neurological Institute Animal Care Committee and conducted according to guidelines and regulations from the Canadian Council on Animal Care (Animal protocol number MNI-8132).

Reviewer #1 (Public review): https://doi.org/10.7554/eLife.101035.3.sa1
Reviewer #2 (Public review): https://doi.org/10.7554/eLife.101035.3.sa2
Author response https://doi.org/10.7554/eLife.101035.3.sa3

## Additional files

### Supplementary files
Supplementary file 1. The list of all probes on the MERFISH panel.

Supplementary file 2. Extended data files contain raw differential expression analysis results in Excel format. Tabs within the file are provided for various comparisons referenced/used throughout the text.

MDAR checklist

### Data availability
Code can be found at https://github.com/AwatramaniLab/Gaertner_Oram_etal (copy archived at *AwatramaniLab, 2025*). MERFISH data has been deposited in Zenodo at https://doi.org/10.5281/zenodo.12636327 and sequencing data have been deposited in GEO under accession code GSE271781.

The following datasets were generated:

| Author(s) | Year | Dataset title | Dataset URL | Database and Identifier |
|---|---|---|---|---|
| Gaerner Z, Oram C, Schneeweis A, Awatramani R, Poulin J | 2024 | Molecular and spatial transcriptomic classification of midbrain dopamine neurons and their alterations in a LRRK2G2019S model of Parkinson's disease | https://www.ncbi.nlm.nih.gov/geo/query/acc.cgi?acc=GSE271781 | NCBI Gene Expression Omnibus, GSE271781 |
| Cameron O | 2024 | MERFISH Dataset of Mouse Dopamine Neurons_03Jul2024 | https://doi.org/10.5281/zenodo.12636327 | Zenodo, 10.5281/zenodo.12636327 |

The following previously published dataset was used:

| Author(s) | Year | Dataset title | Dataset URL | Database and Identifier |
|---|---|---|---|---|
| Gaertner Z, Azcorra M, Awatramani R, Dombeck DA | 2023 | Single nucleus RNAseq of midbrain dopaminergic neuronal nuclei isolated by FACS | https://www.ncbi.nlm.nih.gov/geo/query/acc.cgi?acc=GSE222558 | NCBI Gene Expression Omnibus, GSE222558 |

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
