## [Editor Report · eLife Assessment]

This **important** study combines single nucleus transcriptional profiling with spatial transcriptomics to identify and map heterogeneity among dopamine neurons in the mouse ventral midbrain. The **compelling** results separate dopamine neurons into three broad families that have unique (yet overlapping) spatial distribution within the ventral tegmental area and substantia nigra, and also identify population-specific changes in a LRRK2 mouse model of Parkinson's Disease. The creation of a public-facing app where the snRNA-seq data can be investigated by anyone is a major strength.

---

## [Referee Report · Reviewer #1 (Public review)]

Summary:

Dopamine neurons contribute to motivated and motor behaviors in many ways, and ample recent evidence has suggested that distinct dopamine neuron subclasses support discrete behavioral and circuit functions. Prior studies have subdivided dopamine neurons by spatial localization, gene expression patterns, and physiological properties. However, many of these studies were bound by previous technical limitations that made comprehensive subclassification efforts difficult or impossible. The main goal of this manuscript was to characterize and further define dopamine neuron heterogeneity in the ventral midbrain. The study uses cutting-edge single nucleus RNA-seq (on the 10X Genomics platform) and spatial transcriptomics (on the MERFISH platform) to define dopamine neuron heterogeneity with unprecedented resolution. The result is a convincing and comprehensive subclassification of dopamine neurons into three main families, each with major branches and subtypes. In addition, the study reports comparisons between wild type mice and mice that harbor a G2019S mutation in the Lrrk2 gene, which models a common cause of autosomally dominant Parkinson's Disease in humans. These results, while less robust due to the nature of the group comparisons, nevertheless identify vulnerability within specific dopamine neuron subpopulations. This vulnerability may contribute unique risk to dopamine neuron loss in the context of Parkinson's disease. Overall, the study is careful and rigorous and provides a critical resource for the rapidly evolving knowledge of dopamine neuron subtypes.

Strengths:

-The creation of a public-facing app where the snRNA-seq data can be investigated by anyone is a major strength.

-The manuscript includes careful comparisons to prior datasets that have sought to explore dopamine neuron heterogeneity. The result is a useful synthesis of new findings with previously published work, which is helpful for moving the field forward in this area.

-The integration of snRNA-seq with MERFISH results is particularly strong, and enables insight not only into subclassification, but also into how this relates to spatial localization. The careful neuroanatomy reveals important distinctions between Sox6, Calb1, and Gad2 positive dopamine neuron families, with some degree of spatial overlap.

---

## [Referee Report · Reviewer #2 (Public review)]

Gaertner and colleagues present a study examining the transcriptomic diversity and spatial location of dopaminergic neurons from mice and examine the changes in gene expression resulting from knock in of the Parkinson's LRRK G2019S risk variant. Overall, I found the manuscript presented their study very clearly, well written with very clear figures for the most part. I am not an expert on mouse neuroanatomy but found their classification reasonably well justified and spatial orientation of dopaminergic neurons within the mouse brain informative and clear. While trends were clear and well presented, the apparent spatial heterogeneity suggests that knowledge of the functional connections and roles of these neurons will be required to better interpret the results presented but nonetheless their findings exposed significant detail that is required for further understanding.

The study of the transcriptional effects of the LRRK2 KI was also informative and clearly framed in terms of a focused analyses on the effects of the KI only on dopaminergic neurons.

I thank the authors for addressing my previous concerns and comments, and feel they have done so well. I agree that as GSEA only includes ranked genes from the specific study, the gene set is already limited to the relevant background.

---

## [Author Response]

The following is the authors’ response to the original reviews.

**Reviewer #1 (Public review):**
Weaknesses:(1) Important details about the nature of DEG comparisons between the wild type and the Lrrk2 G2019S model are missing.

Please see the recommendations section below for specific responses to individual comments from Reviewer #1.

(2) Some aspects of the integration between snRNA-seq and MERFISH data are not clear, and many MERFISH-identified cells do not appear to have a high-confidence cluster transfer into the snRNA-seq data space. Imputation is used to overcome some issues with the MERFISH dataset, but it is not clear that this is appropriate.

Please see the recommendations section below for specific responses to individual comments from Reviewer #1.

**Reviewer #2 (Public review):**
(1) In the GO pathway analyses (both GSEA and DEG GO), I did not see a correction applied to the gene background considered. The study focusses on dopaminergic neurons and thus the gene background should be restricted to genes expressed in dopaminergic neurons, rather than all genes in the mouse genome. The problem arises that if we randomly sample genes from dopaminergic neurons instead of the whole genome, we are predisposed to sampling genes enriched in relevant cell-type-specific roles (and their relevant GO terms) and correspondingly depleted in genes enriched in functions not associated with this cell type. Thus, I am unsure whether the results presented in Figures 8 and 9 may be more likely to be obtained just by randomly sampling genes from a dopaminergic neuron. The background should be limited and these functional analyses rerun.

Thank you for pointing out this important concern. We agree that overrepresentation analyses (ORAs) are vulnerable to selecting cell-type specific markers as significantly differentially expressed and thus inflating detection of cell-type associated gene sets rather than those truly altered as a function of experimental condition. We have thus re-run the GO analyses in our study with the genetic background being adjusted for each individual comparison. For dataset-level GO in Fig 8, genetic background was defined as genes with expression detected in at least 5% of all cells (to approximate the inclusion of cluster-specific genes). For comparisons of subsets within the dataset (i.e. a family or cluster) across conditions, a minimum detection level of 10% of cells was used to define the genetic background. These same thresholds were applied to filter the DEG lists used as input for GO. Interestingly, this correction appears to have filtered out or lowered the significance of some of the more generic brain-associated pathways that we initially presented, such as axonogenesis or learning and memory, and we feel even more confident in our original interpretation.

Functional class scoring methods like GSEA, however, are unlike ORAs in that they do utilize a hypergeometric test to calculate overrepresentation as no distinction is made between significant and non-significant differential gene expression (nor is a genetic background provided as input to this tool). GSEA takes as input the full DE results, ranking genes according to their association with either group. Thus, genes simply enriched in DA neurons should be present towards both extremes of the rank list, rather than uniformly skewed toward one extreme. Per the GSEA authors’ user manual and original source paper, the entirety of DE testing should be provided as input for GSEA (barring genes with detection levels so low that their differential expression and/or ranking is likely to be artifactual):

“The GSEA algorithm does not filter the expression dataset and generally does not benefit from your filtering of the expression dataset. During the analysis, genes that are poorly expressed or that have low variance across the dataset populate the middle of the ranked gene list and the use of a weighted statistic ensures that they do not contribute to a positive enrichment score. By removing such genes from your dataset, you may actually reduce the power of the statistic and processing time is rarely a factor as GSEA can easily analyze 22,000 genes with even modest processing power. However, an exception exists for RNA-seq datasets where GSEA may benefit from the removal of extremely low count genes (i.e., genes with artifactual levels of expression such that they are likely not actually expressed in any of the samples in the dataset).” [https://www.gsea-msigdb.org/gsea/doc/GSEAUserGuideFrame.html]

In our study, this filtering of very low expression genes (to account for artifactually inflated fold changes or a large number of ties in the rank list that are subsequently ordered at random) occurred at the level of DE testing using the Seurat FindMarkers command, in which differential expression calculations were only performed for genes that were detected in a minimum of 10% of cells in the dataset.

(2) In the scRDS results, I am unsure what is significant and what isn't. The authors refer to relative measures in the text ("highest") but I do not know whether these differences are significant nor whether any associations are significantly unexpected. Can the x-axis of scRDS results presented in Figure 9 H and I be replaced with a corrected p-value instead of the scRDS score?

An important distinction should be made here between scDRS and similar approaches that utilize overrepresentation analyses to assess for associations of DEGs with putative risk genes, similar to the GO analyses performed in our paper. The scDRS score represents the relative association for each individual cell’s expression profile (among all other cells in the dataset) with PD risk loci by utilizing the underlying SNPs and associations described in GWAS summary statistics (see Methods or Zhang et al., Nat Genetics 2022 for more details). While scDRS can be used to generate a p value for each individual cell in the dataset, scDRS does not have a native method for defining group-level p values, nor have we attempted to calculate group-level p values here. In order to compare cluster-level mean scDRS scores and determine their significance, we created bootstrapped 95% confidence intervals for the mean scDRS score of each cluster or family (shown by the error bars in forest plots 9G, 9H). A score of 0 represents the null hypothesis of no association between gene expression and PD risk loci, and thus if the 95% confidence interval does not overlap 0, the mean scDRS score for a given group can be regarded as significant as there is a less than 5% chance of the true group mean containing the null. Similarly, groups can be compared to each other in the same way to determine if the group-level mean scDRS score is significantly different across a given pair. However, this overlap of confidence intervals should be interpreted cautiously, as there are a large number of potential comparisons that can be made, creating the potential for Type I error. We have added language to clarify what the scDRS score represents, and to ensure it is not conflated with approaches such as GO or GSEA.

(3) The results discussed at the bottom of page 13 [page 14 of new version] state that 48.82% of the proteins encoded by the Calb1 DEGs have pre-synaptic localisations as opposed to 45.83% of the SOX6 DEGs, which does not support the statement that "greater proportions of DEGs are associated with presynaptic locations in cells from vulnerable DA neurons (Sox6 family, [and in particular,Sox6^tafa1]), compared to less vulnerable ones (Calb1 family)".

Thank you for pointing this out; the error here lies in the wording of the results. The percentages mentioned above describe the percentages within the synaptic localized genes rather than the total DEG lists. We have rephrased this section for clarity to include both the percentages within this category as well as the total (the results of which are in line with our original statement).

(4) While an interest in the Sox6^tafa1 subtype is explained through their expression of Anxa1 denoting a previously identified subtype associated with locomotory behaviours, it was unclear to me how to interpret the functional associations made to DEGs in this subtype taken out of context of other subtypes. Given all the other subtypes, it is not possible to ascertain how specific and thus how interesting these results are unless other subtypes are analysed in the same way and this Sox6^tafa1 subtype is demonstrated as unusual given results from other subtypes.

In our study, we chose to specifically focus on this population given its unique acceleration-locked functional activity pattern observed in Azcorra & Gaertner et al, Nat Neuro 2023, as there are technical limitations that warrant cautious application of the above approach. We agree that the associations of this population to the described DEGs cannot be interpreted as unique to this population given the data presented and have added language to this effect within the text. There are two major challenges to analyzing all other subtypes to provide a comparison. Firstly, given the number of subtypes involved and number of downstream analyses, it is computationally intensive to carry out this analysis. More importantly however, the results cannot be easily compared across different populations due to the variability in both cluster size and internal heterogeneity of each cluster, as the statistical power in calculating DEGs will be inherently different across these populations (i.e. smaller or more heterogenous clusters would be expected to show a lower number of DEGs reaching significance). While pseudo bulk testing is effective for mitigating these factors, our limited sample number (n=2 independently generated datasets per group) dramatically underpowers differential expression testing using pseudo bulk analysis. One solution is to uniformly limit each cluster size to the minimally observed cluster size through random down-sampling. While this allows the ‘n’ in DE calculations to be uniform, this potentially worsens the problem of internal heterogeneity, which would remain roughly constant but in the setting of a lower ‘n’, increasing the variability in results for larger clusters. To provide a comparator for the population of interest we focused on, we have performed this down sampling approach in order to compare Sox6^Tafa1 to another cluster within the VTA, Calb1^Stac, that also expresses high levels of *Anxa1* and *Aldh1a1* given the broad interest in these markers as proxies for vulnerability. The results of this comparison are now shown in Figure S10.

(5) On p12, the authors highlight Mir124a-1hg that encodes miR-124. This is upregulated in Figure 8D but the authors note this has been to be downregulated in PD patients and some PD mouse models. Can the authors comment on the directional difference?

We have adjusted the text to reflect this discrepancy and speculate on why this may be observed. In short, one hypothesis is that miR-124, given its proposed neuroprotective effects, is increased in DA neurons facing toxic metabolic insults as a compensatory response. In our prodromal model without observable degeneration, this could represent an early sign of cell stress. While speculative, in PD patients or overtly degenerative models, lack of compensatory miR-124 or fulminant cell death among vulnerable cells could result in an observed decrease in miR-124 expression.

(6) Lastly, can the authors comment on the selection of a LogFC cut-off of 0.15 for their DEG selection? I couldn't see this explained (apologies if I missed it).

The 0.15 cutoff was selected arbitrarily based on the observed range of fold changes seen among our differentially expressed genes. However, importantly, this cutoff was not used for defining DEGs for downstream analyses such as GSEA or GO, nor for defining significance of differential expression, which was done purely based on FDR-adjusted p values <0.05. The selection of 0.15 affects only the coloring seen in the volcano plot, which we have decided to move to supplemental figures given the uniformly small effect size seen in individual genes and a separate reviewer comment regarding concern in the field over differential expression testing methods in single-cell datasets. Instead, this figure now focuses on highlighting pathway- and gene-set level comparisons that can provide easier interpretation of small, but concordant changes across swaths of genes.

**Recommendations for the authors:**

**Reviewer #1 (Recommendations for the authors):**
(1) In the MERFISH dataset, only around half of the DAergic cells (2,297 of 4,532) were successfully projected into the snRNA-seq UMAP space, based on a similarity score > 0.5. Additionally, key transcripts that were used to define the snRNA-seq clusters (such as Sox6) were not identified at all in the MERFISH dataset. This raises some questions about the ability to integrate and compare these datasets directly, which are not fully considered in the manuscript. These discrepancies are smoothed over using imputation, which allows specific class-defining genes such as Sox6 to be plotted on spatial coordinates in Figure 4D. However, imputation is not without caveats, and the appropriateness of the imputation is not well considered in the text.

We fully agree with the reviewer that the use of an imputation approach needs to be clarified and justified thoroughly. We added a sentence to better clarify the process of imputation on Page 9 “The imputed gene expression is extrapolated from anchors established from pairwise correspondences of cell expression levels between MERFISH and snRNA-Seq datasets.” This pair-wise cell correspondence as defined by anchors can be assessed using Seurat confidence score. We acknowledge the fact that only about 50% of cells could confidently be transferred onto the snRNA-Seq data. This is the result of using a stringent confidence level of 0.5 (similar to previous publications, PMID: 38092916 & 38092912). We preferred mapping fewer high-confidence cells than potentially misrepresenting the spatial location of some of these clusters.

It is also important to demonstrate the reliability of gene imputation. Indeed as pointed out by the reviewer, some probes such as *Sox6* were not detected in the MERFISH dataset. To strengthen our data integration and as already mentioned in the manuscript, we excluded 219 genes based on the deviation of average counts per cell between the datasets. The fact that the imputed expression of *Sox6* perfectly reflects its well-characterized distribution (PMIDs: 25127144, 30104732, 25437550, 34758317) strengthened our confidence in our imputation pipeline. We also looked at the correlation of imputed gene expression with the detected transcripts in our MERFISH experiments. We added a new supplemental figure (S7) highlighting the correlations between MERFISH and imputed gene expression of 8 genes (4 for each Sox6 and Calb1 family). Together Fig S6 and S7 show the range of correlations between imputed and actual MERFISH transcript. Altogether, we can observe relatively high correlation between the number of detected transcripts per gene in snRNA-Seq and MERFISH datasets

In addition, we added a paragraph discussing limitations of gene expression imputation on page 17: “A strength of our study is that it utilizes advantages of each transcriptomic approach, the deep molecular profiling of individual cells using snRNA-Seq and the spatial resolution of MERFISH. For instance, we relied on gene expression imputation to ascribe expression level to genes not covered/detected in our MERFISH probe panel. Gene imputation as described by Stuart et al.(92) has been used in several recent studies integrating spatial and transcriptomic data(46, 47). It relies on identifying anchors that enable projection of MERFISH data onto the UMAP space of a snRNA-Seq dataset and then uses neighboring cells to extrapolate the expression of genes not included in our probe panel. This approach was used to impute Sox6 expression, which accurately reflects what has been reported in prior immunofluorescence and in situ hybridization studies(11, 27, 38, 43, 55). Moreover, imputed gene expression levels correlated strongly with MERFISH detected transcript for most genes further supporting our approach (Fig S6 and S7). Nevertheless, dataset integration has limitations that should be considered. First, imputed gene expression relies on the ability to identify reliable anchors linking the snRNA-Seq and MERFISH datasets. These anchors are determined in part by the choice of genes included on probe panels and thus could indirectly influence the reliability of imputed gene expression. Secondly, gene counts per cell in MERFISH are determined via segmentation of images, which is susceptible to artifacts and bias from centrally versus peripherally localized gene transcripts. In summary, although limitations are present in multi-modal transcriptomic analyses, merging these two approaches provided a molecular and spatial map of the DA system that could not have been resolved by either method alone.”

(2) In the discussion, the authors argue that the cellular classifications identified here for DA neurons are more likely to reflect discrete cell types than cell states. The rationale for this conclusion is largely based on the absence of subtype differences between wild-type and LRRK2 G2019S transgenic mice. I do not find this argument to be convincing, because it is still possible that certain subdivisions simply reflect dynamic cell states that are also not grossly altered in the mutant mouse. A stronger argument for this claim would be to include trajectory-based analyses that do not show predicted transition points between nearby or related clusters.

We thank the reviewer for pointing out this particular limitation as differentiating “cell type” and “cell states” been debated in the field for years with no consensus emerging how to address the issue. As suggested, we performed a trajectory analysis using Monocle3 on both control and Lrrk2 samples. We’ve built the trajectory map, taking cluster 20 as the starting node. To avoid potential biased trajectories induced by different cell coverage, we’ve down sampled the Lrrk2 condition to match the number of cells of wildtype. As expected, since most of the DA clusters are not segregated in the UMAP space, the trajectory analysis showed predicted transitions between clusters (see Author response image 1A and 1B). Even though some clusters’ pseudotime score were statistically different between the wildtype and Lrrk2 samples, they overall remained similar (Author response image 1C). This analysis suggests that the LRRK2G2019S mutation induces a mild transcriptional perturbation but does not result in a major cell state drift. Indeed, we believe changes in the observed trajectory path would disappear as the number of cells analyzed increases. Because of this bias introduced by cell coverage, we prefer not to include this trajectory analysis in the manuscript to avoid misleading readers. Thus, as suggested by the reviewer, we softened our claim to “This suggests that our taxonomic scheme is agnostic to a mild perturbation such as LRRK2G2019S, suggesting that our clusters are reflective of cell types, rather than cell states. It is possible that with more severe perturbations, such as a toxin lesion, more substantial alterations of taxonomic schemes are observed(86, 93). However, we expect that for mild insults, day to day behavioral changes, or pharmacological paradigms, our clusters will be resistant to changes, although individual gene levels may vary. Nonetheless, we cannot definitively confirm that a given DA neuron cannot convert from one subtype to another. Ultimately, alternative approaches such as detailed fate mapping of clusters or RNAseq-based trajectory analyses with greater numbers of sampled cells could be used to resolve this question.”.

**Author response image 1. sa3fig1:** (A)Trajectory analysis of wildtype and (B) LRRK2^G2019S^ samples. (C) Pseudotime scores for each cluster across wildtype and Lrrk2 conditions. Error bars represent the confidence of error for false positives discovery rate of 5%.

(3) The relationship between individual samples, GEMwell, and sequenced library should be clarified. If independent samples were combined into one GEMwell, this should be explicitly stated for clarity.

We have revised the text to better clarify the methodology. In brief, each of our 4 independent samples (2 control, 2 mutants; equal sexes per sample) were isolated from n=2 pooled mice (for a total n=8 mice across the 4 samples). Each sample was processed in its own GEM well to produce 4 distinct libraries that were subsequently sequenced and analyzed as described.

(4) Please include more details on DEG testing in the manuscript, this is key for interpreting the robustness of certain findings. Ideally, pseudobulked comparisons would be used here (given concerns in the field that DEG testing where N = number of cells artificially inflates the statistical power, violates assumptions of independence, and results in false positive DEGs).

While we agree that pseudobulk analysis would be ideal for reducing false positives, our study, while exceptionally large in total numbers of DA cells profiled, was generated from 4 total 10X libraries as described above, without any mechanism to definitively demultiplex to the original n=8 source mice. Thus, pseudobulk comparisons would be performed using only n=2 per group, which is below the recommended sample size for these methods. Given this concern, we have moved the volcano plot from Figure 8D to the supplementals and added language to the methods and relevant figure legend acknowledging the limitation in Seurat’s default differential expression analysis methodology.